# THE GENERATION PHASES OF FLOW MATCHING: A DENOISING PERSPECTIVE

## ABSTRACT

Flow matching has achieved remarkable success, yet the factors influencing the quality of its generation process remain poorly understood. In this work, we adopt a denoising perspective and design a framework to empirically probe the generation process. Laying down the formal connections between flow matching models and denoisers, we provide a common ground to compare their performances on generation and denoising. This enables the design of principled and controlled perturbations to influence sample generation: noise and drift. This leads to new insights on the distinct dynamical phases of the generative process, enabling us to precisely characterize at which stage of the generative process denoisers succeed or fail and why this matters.

## 1 INTRODUCTION

Flow matching (FM, Lipman et al., 2023; Albergo & Vanden-Eijnden, 2023; Liu et al., 2023) and diffusion models (Sohl-Dickstein et al., 2015; Ho et al., 2020; Song et al., 2021) have achieved state-of-the-art results in generating images, videos, audio, and even text, where they are able to produce content that is virtually indistinguishable from human-produced ones. Research in the field remains extremely active, with many important questions still open: improving sample quality, making training and inference more efficient (Karras et al., 2022; Rombach et al., 2022) and, perhaps most importantly, understanding why current generative models perform so well.

Despite their striking successes, the precise mechanisms that make current generative models so effective still remain elusive. Identifying those is critical to improve them, and several leads have been proposed: analyzing the behaviour of the generative process across time (Biroli et al., 2024), exploiting connections with the exact minimizer of the training loss (Kamb & Ganguli, 2025; Niedoba et al., 2025), explaining memorisation and generalisation (Kadkhodaie et al., 2024; Sclocchi et al., 2025), or the impact of optimization procedures (Wu et al., 2025). Some of these studies, of theoretical nature, have provided elegant interpretations (e.g. the "target stochasticity" of Vastola, 2025, to explain generalisation), but were later questioned by empirical findings (Bertrand et al., 2025). **This highlights the need for carefully designed empirical frameworks that can probe such theories** and, in doing so, guide the development of better methods through deeper theoretical understanding.

In this work, we aim to bring new elements of understanding to the behaviour of flow matching by exploiting a denoising perspective. To this end, we construct a toolkit of denoisers, which differ in their parametrizations and the weighting schemes applied to their training losses. This perspective allows us to directly relate denoising performance to generative performance. By leveraging the equivalence between learning the ideal velocity field in flow matching and learning an ideal denoiser at each time step, we use this toolkit to address the following questions: Is a good generative model essentially nothing more than a good denoiser at every noise levels? Are there specific times during the generative process where accuracy matters most? How do early versus intermediate phases of generation contribute to generalisation and sample quality?

To answer these questions, our contributions are:

1. We design a *denoising toolkit*, namely, controlled procedures to test the impact of several factors on the performance of flow matching models. We show that different denoising losses and parametrizations, though theoretically equivalent if perfectly trained, lead to very different empirical performance, where denoising and generation quality are strongly related.

2. We engineer two types of controlled perturbations applied on the generation process: drift- and noise-type perturbations. We show that they impact distinct temporal phases of generation. We further analyze

these generation stages by exhibiting a discrepancy between the spatial regularities of target and learned velocity fields, *at early times*.

3. We exploit the generality of the denoising framework, that allows building new models in a principled manner, with similar generation performance but different generation behaviours. We highlight the importance *of the intermediate stage* for generation by learned models, a stage that is not revealed by the closed-form optimal velocity.

The structure of the paper is the following: Section 2 provides an introduction to flow matching and related works; Section 3 lays down the theoretical equivalence between flow matching and denoising, which Section 4 empirically attests. Section 5 leverages our framework to design relevant modifications of the generative process. Finally, Section 6 provides new insights on the phases of generation and their nature.

## 2 PRELIMINARIES

**Background on flow matching**  The generative process is defined over time $t \in [0, 1]$, with an initial sample $x_0 \sim p_0$ and a target sample $x_1 \sim p_1$. To connect with the concept of denoisers in the sequel, we further assume that the latent distribution is standard Gaussian: $p_0 = \mathcal{N}(0, \mathrm{I}_d)$, and we work in the setting where the coupling $p_{(x_0,x_1)}$ is the product coupling $p_0 \otimes p_1$. In flow matching, generation of new samples is performed via the numerical resolution of an ordinary differential equation (ODE) on $[0, 1]$:

$$\begin{cases} x(0) = x_0 \sim p_0 \\ \dot{x}(t) = v(x(t), t) \; \forall t \in [0, 1] \,, \end{cases} \tag{1}$$

the function $v : \mathbb{R}^d \times [0, 1] \to \mathbb{R}^d$ being called the *velocity*. The generated sample is simply the ODE solution at time $t = 1$, namely $x(1)$; for an appropriate velocity, it should behave like a sample from $p_1$. In practice, the velocity is parametrized by a neural network $v_\theta$ and learned by solving:

$$\min_\theta \; \mathbb{E}_{t \sim \mathcal{U}[0,1], \, x_0 \sim p_0, \, x_1 \sim p_1} \left[ \| v_\theta(x_t, t) - (x_1 - x_0) \|^2 \right], \tag{2}$$

where $x_t := (1 - t)x_0 + tx_1$ is the linear interpolation between $x_0$ and $x_1$. It is well-known that the solution $v^\star$ to this problem (over all measurable functions) is given by a conditional expectation:

$$v^\star(x_t, t) = \mathbb{E}[x_1 - x_0 \mid x_t, t]. \tag{3}$$

In practice, sampling from $p_1$ in (2) is impossible, and points $x_1$ are instead drawn from a dataset $x^{(1)}, \ldots, x^{(n)}$ of samples from $p_1$. Effectively, $p_1$ is replaced in (2) by the empirical measure $\hat{p}_1 = \frac{1}{n} \sum_{i=1}^n \delta_{x^{(i)}}$. This has an important consequence: the minimizer of (2), when $p_1$ is replaced by $\hat{p}_1$, admits a closed-form $\hat{v}^\star$ (Gu et al., 2025; Bertrand et al., 2025):

$$\hat{v}^\star(x, t) = \sum_{i=1}^n \lambda_i(x, t) \frac{x^{(i)} - x}{1 - t}, \quad \text{where} \quad \lambda_i(x, t) = \mathrm{softmax}\left( \left( -\frac{\|x - tx^{(j)}\|^2}{2(1-t)^2} \right)_j \right)_i. \tag{4}$$

Paradoxically, using this closed-form velocity for generation can only reproduce samples from the training set. At the same time, this target velocity exhibits different behaviours across time: at $t = 0$, it points towards the dataset mean, while as $t \to 1$ it points towards a single training point. Thus, understanding why trained FM models work comes down to determine, qualitatively and quantitatively, at which times $t$ and points $x$ the models must deviate from the closed-form in order to generate new samples, still consistent with the data distribution.

**Related works**  First, flow matching with Gaussian $p_0$ is known to be equivalent to diffusion, for which there is an abundant literature about weighting the loss function across time (Hang et al., 2023; Kingma & Gao, 2023); in diffusion, Kumar et al. (2025) lays down the different losses (noise prediction, score prediction, etc) and the loss weighting they induce. Leclaire et al. (2025) interprets diffusion as iterative "Noising-Relaxed Denoising" to better understand the noise schedules. Several works highlight the existence of distinct phases in the generative process (Biroli et al., 2024; Bertrand et al., 2025), which Sclocchi et al. (2025) connect to learning of high- and low-level features; a generalisation followed by a memorisation phase also seems to appear in training (Bonnaire et al., 2025). A line of work seeks to understand the reasons for generalisation in generative models, characterizing the velocities/scores that are actually learned (Kadkhodaie et al., 2024; Kamb & Ganguli, 2025; Niedoba et al., 2025). We refer the reader to Appendix A for a detailed discussion of related works.

## 3 EQUIVALENCE BETWEEN FLOW MATCHING AND DENOISING

### 3.1 FROM FLOW MATCHING TO DENOISING

We now lay down the procedure to construct a denoiser from a flow matching model. Using the identity $x_t = (1 - t)x_0 + tx_1$ and the expression of the optimal velocity field (3), ones obtains the denoising identity[1]

$$D_t^\star(x_t) := \mathbb{E}[x_1|x_t, t] = x_t + (1 - t)v^\star(x_t, t), \quad (5)$$

namely the Minimum Mean Square Error estimator of the clean image $x_1$ given the noisy observation $x_t$ at time $t$. Thus, any optimally trained FM model naturally yields an optimal denoiser. Building on (5), several works on flow-matching for image restoration define a denoiser at time $t$ as $D_t : x_t \mapsto x_t + (1 - t)\, v(x_t, t)$, see e.g. Zhang et al. (2024); Pokle et al. (2024); Martin et al. (2025).

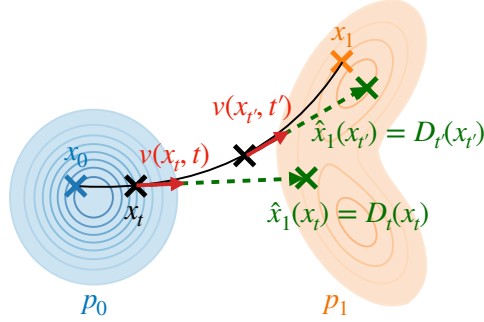

Figure 1: Equivalence between velocity $v_t$ and denoiser $D_t$. Learning the optimal velocity amounts to learning an optimal denoiser at every time $t$.

### 3.2 FROM DENOISERS TO VELOCITIES: THE DENOISING TOOLKIT

The same way a velocity field can be mapped to its associated denoiser, any denoiser $D$ induces a velocity field $v(x, t) = \frac{D_t(x) - \mathrm{Id}}{1 - t}$. This duality yields the question at the heart of our study: *is flow matching nothing more than learning a denoiser at all possible noise levels, and then sampling by following the velocity derived from it?*

To investigate this, we propose a systematic way of constructing denoisers. Throughout the paper, we use the term *denoiser* in a broad sense: a function that maps a noisy input, obtained by corrupting $x \sim p_1$, to a clean estimate of $x$. We will consider *generative denoisers* $D_t$, defined for each $t \in [0, 1]$ and taking as input images of the form $x_t = tx + (1 - t)\varepsilon, \varepsilon \sim \mathcal{N}(0, I_d)$.

**Remark 1** (Equivalence of generative and classical denoisers). Generative denoisers differ from *classical denoisers* $D_\sigma$, parameterized by a noise level $\sigma$ and taking inputs of the form $x_\sigma = x + \sigma\varepsilon$. These two forms are equivalent up to rescaling. If the denoiser is parameterized by a noise level but the input is given as $x_t = tx + (1 - t)\varepsilon$, one can define $x_\sigma = \frac{x_t}{t}$ and apply the classical denoiser $D_\sigma$ with $\sigma = \frac{1-t}{t}$ to approximately recover $x$. Conversely, if the input is of the form $x_\sigma = x + \sigma\varepsilon$, setting $\tilde{x} = \frac{x_\sigma}{1+\sigma}$ and $t = \frac{1}{1+\sigma}$, a generative denoiser $D_t$ can be applied to $\tilde{x}$ to recover $x$. The mapping from $\sigma$ to $t$ is a bijection between $[0, \infty)$ and $(0, 1]$.

Equipped with this definition, we introduce a *denoising toolkit*, a family of neural denoisers obtained by numerically solving optimization problems of the form

$$\underset{D \in \mathcal{C}}{\text{minimize}}\ \mathcal{L}(D), \quad (6)$$

where $\mathcal{C}$ is the class of functions parameterized by a neural network, and $\mathcal{L}(D)$ is the training loss. For specific $\mathcal{C}$ and $\mathcal{L}$, we recover the standard flow matching model, but making these design choices explicit and decoupling them, this abstraction opens the way to a systematic study of their impact.

### 3.3 DENOISING LOSSES $\mathcal{L}$

We present three types of denoising losses, all expressed for a *generative denoiser* $D : \mathbb{R}^d \times [0, 1] \to \mathbb{R}^d$, taking as input $x_t = (1 - t)x_0 + tx_1$, with clean image $x_1 \sim p_1$ and noise $x_0 \sim \mathcal{N}(0, I_d)$.

**Flow matching denoising loss.** The first loss arises directly from the flow matching objective in Eq. (2). Substituting the velocity $v(x, t) = \frac{D(x,t) - x}{1 - t}$ into Eq. (2) together with $x_1 - x_0 = \frac{x_1 - x_t}{1 - t}$, one can write the standard FM objective as a function of the denoiser,

$$\mathcal{L}_{\text{FM}}(D) := \mathbb{E}_{\substack{t \sim \mathcal{U}[0,1] \\ x_0 \sim \mathcal{N}(0, I_d) \\ x_1 \sim p_1}} \left[ \frac{1}{(1 - t)^2} \|D(x_t, t) - x_1\|^2 \right]. \quad (7)$$

---

[1]the denoiser is interchangeably written $D(x, t)$ or $D_t(x)$ for concision

Table 1: Summary of the denoising losses (left) and parametrization classes (right).

| Losses $\mathcal{L}$ | Parametrization classes $\mathcal{C}$ |
|---|---|
| $\mathcal{L}_{\text{FM}} : w_t^{\text{FM}} = \frac{1}{(1-t)^2}$ | $\mathcal{C}_{\text{NN}}: D(x,t) = N^\theta(x,t)$ |
| $\mathcal{L}_{\text{classic}} : w_t^{\text{classic}} = \mathbf{1}_{[(1+\sigma_{\max})^{-1},1]}(t) \cdot t^{-2}$ | $\mathcal{C}_{\text{I+NN}}: D(x,t) = x + (1-t)N^\theta(x,t)$ |
| $\mathcal{L}_{\text{den}} : w_t^{\text{den}} = 1$ | |

Thus, training a denoiser under FM amounts to minimizing a weighted Mean Squared Error (MSE), where the error at time $t$ is weighted by $w_t^{\text{FM}} := (1-t)^{-2}$.

**Classical denoising loss.** As we have seen, classical denoisers are parameterized by a noise level $\sigma$ and trained on inputs $x_\sigma = x_1 + \sigma x_0$. Such a denoiser $\tilde{D}$ is usually trained on noise levels ranging from 0 to $\sigma_{\max}$, by minimizing

$$\mathcal{L}(\tilde{D}) = \mathbb{E}_{\substack{\sigma \sim \mathcal{U}([0,\sigma_{\max}]) \\ x_0 \sim \mathcal{N}(0,I_d) \\ x_1 \sim p_1}} \left[ \|\tilde{D}(x_\sigma, \sigma) - x_1\|^2 \right]. \tag{8}$$

Using the equivalence between the $\sigma$- and $t$-parameterizations (Remark 1) and the change of variables $\sigma = \frac{1-t}{t}$, one can rewrite this loss as

$$\mathcal{L}_{\text{classic}}(D) := \mathbb{E}_{\substack{t \sim \mathcal{U}([1/(1+\sigma_{\max}),1]) \\ x_0 \sim \mathcal{N}(0,I_d) \\ x_1 \sim p_1}} \left[ \frac{1}{t^2} \|D(x_t, t) - x_1\|^2 \right]. \tag{9}$$

Compared to the FM loss $\mathcal{L}_{\text{FM}}$, classical denoising therefore differs in two important ways: *(i)* the loss includes a weight $w_t^{\text{classic}} := \mathbf{1}_{[(1+\sigma_{\max})^{-1},1]}(t) \cdot t^{-2}$; *(ii)* the range of $t$ is truncated to $[1/(1+\sigma_{\max}), 1]$, which covers the full interval $[0,1]$ only if $\sigma_{\max} = \infty$. In other words, classical denoisers cannot handle very low SNR regimes except if trained with unbounded noise levels. In practice, we set $\sigma_{\max} = 19$ so that[2] $t_{\min} = 1/(1+\sigma_{\max}) = 0.05$.

**Unweighted denoising loss.** A natural comparison baseline is to use "no weights" and train with the plain mean squared error (i.e., weighting $w_t^{\text{den}} := 1$):

$$\mathcal{L}_{\text{den}}(D) := \mathbb{E}_{\substack{t \sim \mathcal{U}[0,1] \\ x_0 \sim \mathcal{N}(0,I_d) \\ x_1 \sim p_1}} \left[ \|D(x_t, t) - x_1\|^2 \right]. \tag{10}$$

**More general weightings in denoising losses.** The three losses considered above emphasize opposite time intervals: $\mathcal{L}_{\text{FM}}$ stresses large $t$ (lightly corrupted inputs), while $\mathcal{L}_{\text{classic}}$ stresses small $t$ (highly corrupted inputs). This is expressed by the time-dependent weightings $w_t$ in the loss $\mathbb{E}_{t,x_0,x_1} \left[ w_t \|D(x_t,t) - x_1\|^2 \right]$. As part of our methodology, in Section 6 we will also explore handcrafted weightings, allowing us to probe the generation process. Kingma & Gao (2023) provide similar derivations and additional weightings.

### 3.4 EQUIVALENCE IN THE OPTIMAL SETTING

A critical point is that all the above losses *share the same minimizer* in $L^2(\mathbb{R}^d \times [0,1])$, namely the MMSE denoiser $D^\star(x_t, t) = \mathbb{E}[x_1 | x_t, t]$. However, in practice, minimization is restricted to a parametric function class $\mathcal{C}$, optimization algorithms may be imperfect, and thus different loss weightings can lead to different numerical solutions.

### 3.5 PARAMETRIZATIONS OF THE DENOISERS AND VELOCITIES

We now describe two parametrization classes $\mathcal{C}$ for the denoisers, which are used in the minimization of the losses $\mathcal{L}$ defined in the previous subsection. In all cases, $N^\theta$ denotes a neural network, with parameters $\theta$ belonging to some set $\Theta$.

**Class $\mathcal{C}_{\text{NN}}$: Standard neural network parametrization.** A straightforward approach is to directly parametrize $D$ by a neural net $N^\theta$ taking as input the noisy image and the time:

$$\mathcal{C}_{\text{NN}} = \left\{ D : \mathbb{R}^d \times [0,1] \to \mathbb{R}^d \mid D : x, t \mapsto N^\theta(x,t), \theta \in \Theta \right\}. \tag{11}$$

---

[2]In traditional denoising, models are usually trained with noise level at most $\sigma = 100/255 \simeq 0.4$.

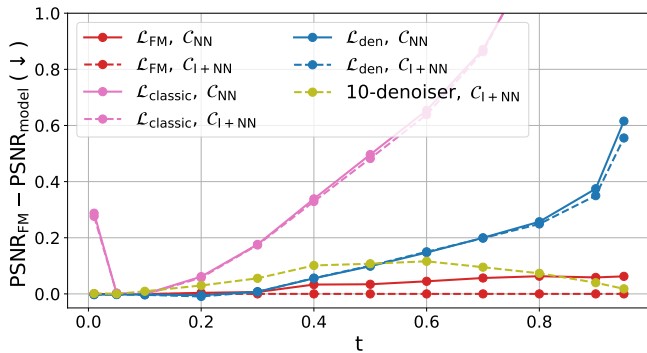
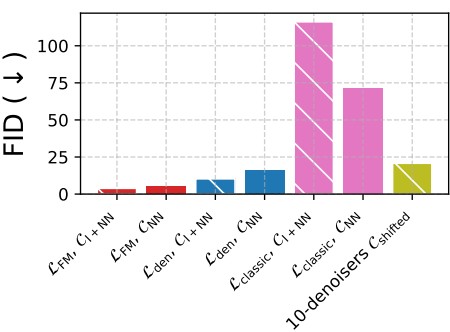

(a) Difference in PSNR (lower is better) between standard FM ($\mathcal{L}_{\mathrm{FM}}, \mathcal{C}_{\mathrm{I+NN}}$) and various models, computed on 1000 test images. Positive values indicate worse denoising performance compared to standard FM.

(b) FID on 50k train images (lower is better).

Figure 2: PSNR and FID for the different losses and parametrizations, CIFAR-10. Models that reach the highest PSNR (low difference in PSNR compared to standard FM) also reach the lowest FID.

**Class $\mathcal{C}_{\mathrm{I+NN}}$: Residual denoiser form $D = \mathrm{Id} + (1-t)N^{\theta}$.** Following the relationship between the optimal FM denoiser $D^{\star}$ and the optimal velocity field $v^{\star}$, namely $D^{\star} = \mathrm{Id} + (1-t)v^{\star}$, one can also parametrize the denoiser in residual form, where the network acts as a correction to identity:

$$\mathcal{C}_{\mathrm{I+NN}} = \left\{ D : \mathbb{R}^d \times [0,1] \to \mathbb{R}^d \mid D : x, t \mapsto x + (1-t)N^{\theta}(x,t), \theta \in \Theta \right\}. \quad (12)$$

A key feature of this parametrization is that it enforces at $t = 1$, $D_t = \mathrm{Id}$, meaning the denoiser leaves its input unchanged – which is the expected behavior (no noise in the input). It also matches the parametrization of standard flow matching models, where the velocity field is directly parametrized by a neural network. We can already hypothesize that *this implicit bias* may benefit the model. The different losses and parametrizations considered are summarized in Table 1.

## 4 GENERATION: DENOISING AT EVERY TIME LEVEL?

### 4.1 DENOISING AND GENERATIVE METRICS

As a first investigation, we evaluate models trained using the different couples of denoising losses/parametrizations $(\mathcal{L}, \mathcal{C})$ on CIFAR-10 (32×32, Krizhevsky & Hinton, 2009) and CelebA-64 (64×64, Yang et al., 2015). All models share the same architecture and training hyperparameters, borrowed from the standard FM training (full details in Appendix C). We also train 10 independent denoisers with $\mathcal{L}_{\mathrm{den}}$, each trained only on the time interval $[i/10, (i+1)/10]$ for $i \in [\![0,9]\!]$, yielding a velocity field defined in a piecewise manner over time; the resulting model is denoted "10-denoisers".

Figure 2 displays the performance of our trained models both in denoising and in generation. Denoising at noise level/time $t$ is evaluated using the Peak Signal-to-Noise Ratio (PSNR) between denoiser output $D_t(x_t)$ and clean sample $x_1 \sim p_1$, averaged on 1000 test images. Generation is evaluated using the Fréchet Inception Distance (FID, Heusel et al., 2017) between 50k train samples and 50k generated samples. First, Figure 2 confirms that, although all denoising losses $\mathcal{L}$ are equivalent in terms of the optimal denoiser they promote, both **the choices of $\mathcal{L}$ and $\mathcal{C}$ impact performance** in practice[3]. For every loss except the $\mathcal{L}_{\mathrm{classic}}$ (for which the generated images are of such poor quality that the corresponding FID values are not meaningful), the residual parametrization $\mathcal{C}_{\mathrm{I+NN}}$ consistently outperforms the plain parametrization $\mathcal{C}_{\mathrm{NN}}$, supporting the hypothesis that explicitly enforcing $D_t = \mathrm{Id}$ introduces a beneficial implicit bias. Second, **the models with lower FID also have better PSNRs** at all noise levels, except for 10-denoisers which has worse PSNR than standard flow matching for $t < 0.9$. The best performance is obtained with the FM loss under the parametrization $\mathcal{C}_{\mathrm{I+NN}}$ (i.e. the standard flow matching approach). On the contrary, the models trained with the classical loss ($w_t^{\mathrm{classic}} = t^{-2}$) and the unweighted denoising

---

[3]We provide in Appendix D a table with additional tested weightings, showing the same trends.

loss ($w_t^{\mathrm{den}} = 1$), both motivated only by denoising considerations, obtain not only poorer generation performance, but also poorer denoising performance. Combined with the good performance of the 10-denoisers, this suggests that training a *single* network (taking $t$ as a parameter) to denoise at every noise level, if not counterbalanced with an appropriate use of weights, is detrimental to performance. Perhaps surprisingly, the FM weights $w_t^{\mathrm{FM}} = 1/(1-t)^2$ turn out to be the most efficient for denoising, although they put more emphasis on accurate denoising at low noise level ($t$ close to 1), which one may think of as an easy task.

On top of these general trend, the behaviour of 10-denoisers is more complex: although it yields worse PSNRs than the FM denoiser (except at $t \geq 0.9$), it still manage to reach a comparable FID.

## 4.2 INPAINTING

We complement our experiments with a third metric, at the crossroads of denoising and generation, to confirm the correlation previously observed. To this end, we turn to an *intermediate task*: image inpainting. This task is both an inverse problem, a field in which denoisers are regularly used in a plug-and-play fashion (Venkatakrishnan et al., 2013; Meinhardt et al., 2017), and a generative challenge, as the model must synthesize large missing parts of the image. We evaluate our denoisers in this setting using the PnP-Flow algorithm (Martin et al., 2025), which incorporates FM models into plug-and-play frameworks. This method builds on the equivalence between denoisers and FM velocities, since in PnP-Flow the time-dependent denoiser is directly induced by the learned velocity field. Experimental details and visual examples are provided in Appendix F, Figure 10. As shown in Figure 3, the ranking of models on the inpainting task coincides with their ranking in terms of FID and PSNR. Now that it is clear that the considered models are uniformly good or bad across all three metrics, we investigate in more depth the factors that may cause this performance gap.

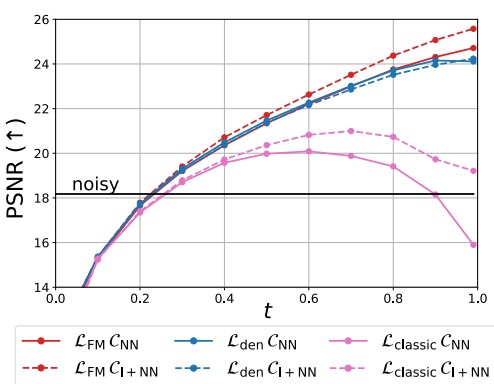

Figure 3: Inpainting results in terms of PSNR (higher is better) as a function of the time in PnP-Flow, CelebA-64. Results are averaged over 100 images. Mask of size $17 \times 17$. The horizontal black line represents as a reference the PSNR of the degraded image.

## 5 PERTURBATING THE GENERATION PROCESS

The above experiments revealed striking differences of performance between models trained with different losses; naturally raising the question: why are some of the considered models much worse at generating than the FM baseline and at which stage of the generative process do they fail? A first step towards addressing this question is to develop a more fine-grained understanding of the temporal structure underlying the generation process. To do so, we implement controlled perturbations of the denoiser, applied at different selected time intervals during the generation process.

More precisely, from the standard FM denoiser $D_t$ (trained with $\mathcal{L}_{\mathrm{FM}}, \mathcal{C}_{\mathrm{I+NN}}$) we create controllable perturbed denoisers, equal to $D_t$ everywhere except on a given time interval $[t_{\min}, t_{\max}]$, where they are equal to $\tilde{D}_t \triangleq D_t + \sigma(t)\delta$. The controllable factors are:

1. the perturbation interval $[t_{\min}, t_{\max}]$. We consider intervals of length 0.3;

2. the level of perturbation $\sigma(t)$. We set it such that $\tilde{D}_t$ has a PSNR equal to 90% of that of $D_t$;

3. the (deterministic) perturbation direction $\delta$. We evaluate both *low- and high-frequency perturbation patterns*. We consider *(a)* checkerboard perturbations corresponding to alternated patches of $+1$ and $-1$, with different patch sizes; *(b)* positive (resp. negative) shift respectively referring to a constant perturbation of $+1$ (resp. $-1$); *(c)* "Residual" referring to the perturbation $\tilde{D}_t := D_t + \sigma(t)(\mathrm{Id} - D_t)$ which can be seen as a relaxed denoiser. (d) low-pass–filtered Gaussian perturbations, corresponding to $\delta = h \star g_0$ where $g_0$ is a fixed Gaussian noise realization and $h$ is a low-pass filter with various cutoffs. Perturbations are displayed in Appendix G, Figures 12 and 13.

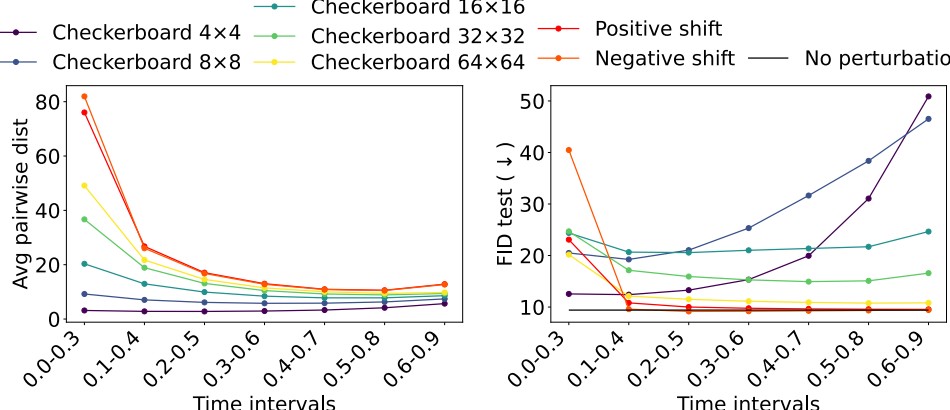

Figure 4: Influence of different perturbations at different generation phases on the FID (10K, test) on CelebA 128. Two classes of perturbations emerge: **high-frequency perturbations** (ckb. 4x4, 8x8) characterized by high FID, low pairwise distance and strongest impact in the last times and **low-frequency perturbations** (pos./neg. shift, ckb. 16x16-64x64) characterized by low FID, high pairwise distance and strongest impact in the early times.

Results are presented Figure 4 and in Appendix G. The left plot displays the average $\ell_2$ distance between $500$ generated samples and their corresponding baseline samples (i.e., generated with $D_t$ starting from the same noise instance $x_0$). This allows to measure how the injected noise deviates the ODE trajectory and affects the generated sample. The right plot displays the test FID-10k for each perturbation applied on each interval.

We observe that the $1 \times 1$ checkerboard perturbation does not induce any significative change in the generated samples, meaning it is effectively corrected after the perturbed time interval (although, by design of the experiment, like all perturbations it degrades the PSNR by 10 %). As the size of the patches grows from $1 \times 1$ to $16 \times 16$, the generated images deviate further from the initial samples. Something surprising happens: although larger patches always induce higher drift in the distribution (as assessed by the pairwise distance), we observe that applying the $4 \times 4$ kernel size at later times produces a large increase in FID, while for the other checkboard perturbations, they remain quite low. Regarding the constant perturbations, they strongly alter the generated images, producing washed-out outputs (see Appendix G, Figures 14, 15 and 18) but yield only a small FID change. On the contrary, the residual perturbation leads to a strong increase of the FID, although the generated images are noisy versions, close to the initial ones in $\ell_2$ distance and visually.

These findings lead us to the following takeaways:

1. **First, despite all perturbations having, by design, the same impact on PSNR degradation, they do not affect FID in the same manner.** A first remark is that it is possible to build denoisers with degraded denoising performance that still remain good generators. Second, the experiment shows a stronger sensitivity to *high-frequency perturbations* (i.e. low patch-size checkerboard pattern or residual, both being Gaussian-like) than to *low-frequency perturbations* (i.e. positive shift, negative shift, large patch-size checkerboard).

2. Second, **low-frequency perturbations**, which induce a drift in the sampled distribution (i.e. a global change to the full image) **are more impactful when applied early**. In contrast, **high-frequency perturbations,** which produce noise-like local changes, **have a stronger effect when applied in late time intervals**. This aligns with previous studies on the effective receptive field of the velocity U-Net during generation: Kamb & Ganguli (2025, Fig. 4.a) and Niedoba et al. (2025, Fig. 3) show that it evolves from a large kernel that encompasses the full image (enabling the model to compute the average of the dataset at $t = 0$, matching the closed-form target) to a local field (a few pixels) for the last time steps, corresponding to removal of very small noise.

| No perturb. | | Low-frequency | | High-frequency | |
| --- | --- | --- | --- | --- | --- |
| | **Interval** | Neg. shift | Ckb. 64×64 | Ckb. 8×8 | Ckb. 4×4 |

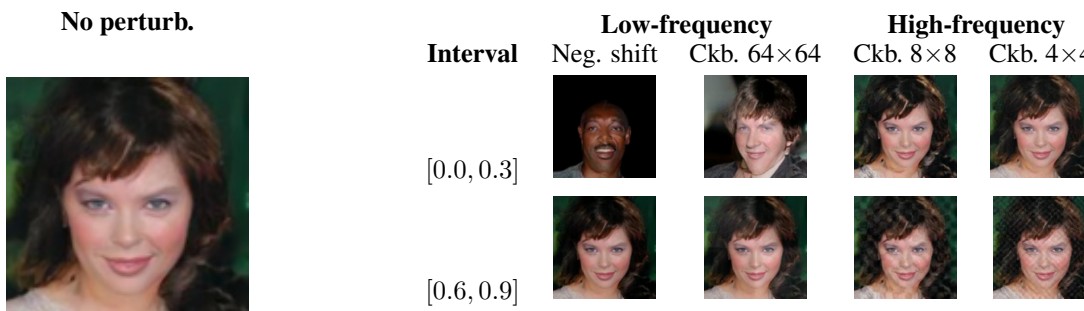

Figure 5: Effect of low- and high-frequency perturbations at early and late times for CelebA 128.

# 6  THE DISTINCT INFLUENCES OF EARLY AND LATE TIMES ON GENERATION

Previous experiments showed the influence of the early phase of the generative process, sensitive to low-frequency perturbations with a drift effect on samples and late phase, sensitive to high-frequency perturbations with a noise effect on samples. This distinction of small and large times has already been observed in different contexts: Biroli et al. (2024) identify different temporal regimes in the generative process, evolving from trajectories that are indistinguishable to trajectories that all converge toward the training dataset, thereby revealing a phase transition between generalisation and memorisation. Bertrand et al. (2025) show that learned velocities differ the most from the loss minimizer $\hat{v}^\star$ (4) at small and large $t$, and that small $t$ seem to be more important for creating new images.

## 6.1  REGULARITY DISCREPANCIES BETWEEN TARGET AND LEARNED MODELS

To deepen our understanding of the various phases, we consider a simpler setup with $p_0$ the uniform distribution on $[-1, 1]^d$, and the discrete target distribution $\hat{p}_1$. In this setting, the optimal velocity field $\hat{v}^\star$ admits a simple closed-form expression (Bertrand et al., 2025, Prop. 1). It is defined on cones [4] $C^{(i)} = \{x \in \mathbb{R}^d : \exists t \in [0, 1], x_0 \in [-1, 1]^d, x = (1-t)x_0 + tx^{(i)}\}$. The optimal velocity $\hat{v}^\star(x_t, t)$ simply points towards the mean of training points $x^{(i)}$ for all the indices $i$ such that $x_t \in C^{(i)}$ (Bertrand et al., 2025, Prop. 1). It follows that for $t$ close to 0, $\hat{v}^\star(x_t, t)$ points towards the mean of the dataset, whereas in the later steps, $x_t$ belongs to a single cone and $\hat{v}^\star(x_t, t)$ points towards the associated data point (Figure 6). In between, there is a transition phase when the trajectories "split" into different cones.

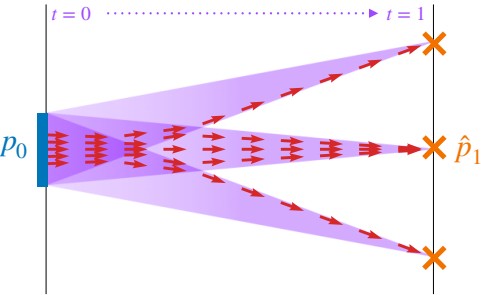

Figure 6: Splitting trajectories in 1D flow matching between a uniform $p_0$ and a discrete target $\hat{p}_1$, composed of 3 data points. Red arrows represent the optimal velocity $\hat{v}^\star(x_t, t)$.

This setting examplifies distinct phases of the target closed-form velocity/denoiser plotted on Figure 7a:

- *Very early phase.* The closed form denoiser outputs the mean of the dataset: this is the best estimate at very high noise levels. Since the target denoiser is constant at $t = 0$, its Jacobian norm is zero at these initial times.

- *Early phase.* The closed-form velocity/denoiser rapidly changes, visible as a peak in the Jacobian norm. This corresponds to the splitting of ODE trajectories.

- *Intermediate phase.* After some time threshold $\tau$, points $x_t$ only belong to a single cone $\mathcal{C}^{(i)}$ and the velocity, equal to $\frac{x^{(i)} - x}{1-t}$, varies very smoothly. Equivalently, the denoiser's output is constant equal to $x^{(i)}$ and its Jacobian norm vanishes.

- *Late phase.* Due to the factor $1/(1-t)$ in the velocity formula, the Jacobian norm of the target velocity explodes while the target denoiser's one remains null.

---

[4]For a Gaussian $p_0$, the optimal velocity $\hat{v}^\star_t$ is defined on the whole space: the regions not covered by cones in the uniform case correspond to regions of very low probability in the Gaussian setting.

We conjecture that approximating the closed-form is the hardest at such critical $\tau$ because of a high local Lipschitz constant of the target velocity field with respect to $x$, making it difficult to accurately capture the trajectory splitting dynamics. We also conjecture that it is desirable to maintain a higher Lipschitz constant that the target velocity / denoiser in the intermediate phase.

To test this hypothesis numerically, on Figure 7b, we estimate the local Lipschitz constant of the velocity in $x$ at time $t \in [0, 1]$ by computing the spectral norm of the Jacobian $\nabla_x v(x_t, t)$ along sampled ODE trajectories, using the power method, both for the closed-form $\hat{v}^\star$ and for velocities induced by our set of denoisers. This quantity reflects how strongly trajectories diverge locally.

Once again, we identify distinct temporal behaviours. First, *in the early phase ($t \in [0.1, 0.3]$)*, the closed-form velocity exhibits a sharp Lipschitz peak, while our trained models, by contrast, fail to reproduce this peak; which confirms our intuition. The pronounced gap - at a phase already identified as critical for generalisation (Bertrand et al., 2025) - shows that networks learn a smoother velocity, unable to match the closed-form, and that this *actually helps* trajectories drift away from training samples thereby favoring generalisation. Second, *in the intermediate regime ($t \in [0.3, 0.8]$)*, the models maintain a relatively high Jacobian spectral norm, consistently above the closed-form, with the best-performing models showing the largest values in this range. This shows that maintaining a high Lipschitz constant during intermediate times is not problematic; instead, it may reflect the capacity of neural networks to represent complex transformations (Salmona et al., 2022).

While only early times are critical when considering the closed-form (time when ODE trajectories split, inducing a peak in the Lipschitz constant), our experiment suggests that, when it comes to learned models, **additional factors governing good generation may still occur later**. We explore this intermediate phase further in the next section.

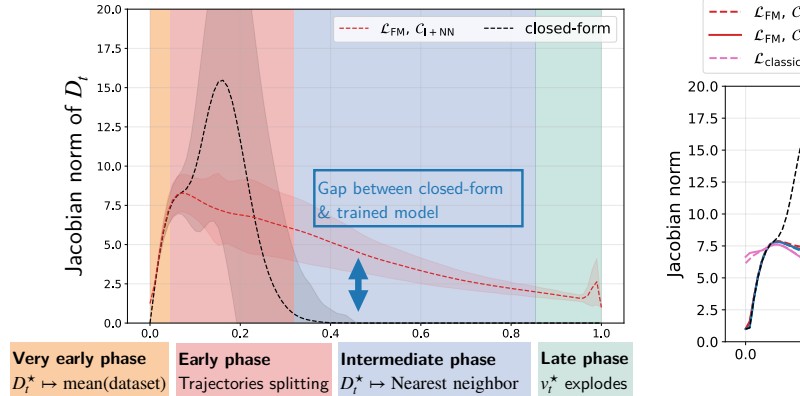
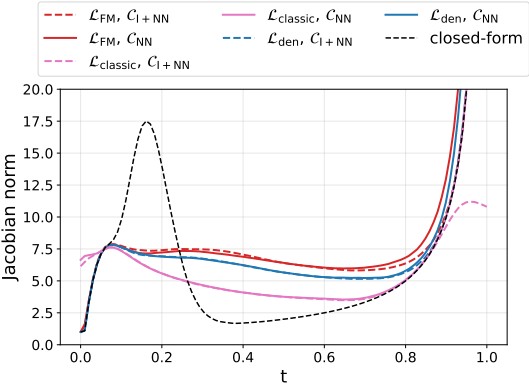

(a) Mean of the spectral norm $\|\nabla_x D(x_t, t)\|_2$ computed on 1000 ODE trajectories on Cifar10 with Gaussian $p_0$. Colored regions indicate the temporal phases defined by the closed-form dynamics.

(b) Mean of the spectral norm $\|\nabla_x v(x_t, t)\|_2$ computed on 1000 ODE trajectories on Cifar10 with Gaussian $p_0$.

Figure 7: The closed-form velocity shows a transition when trajectories split, producing an early peak in the Lipschitz constant. The trained models do not exhibit such distinction, and **learn a smoother field**.

## 6.2 PROBING THE TEMPORAL PHASES WITH AD-HOC DENOISERS

In Section 5, we investigated how to artificially perturb a pretrained denoiser, identifying two types of behaviors: early-time drifts and late-time FID degradation. We now ask whether these effects can be reproduced directly through training, by modifying either the loss function or the class of functions over which the loss is minimized, while remaining within the framework of our denoising toolkit.

In order to explore the importance of matching the Lipschitz peak at $t = 0.2$ of the closed form (see Figure 7b), we build a denoiser whose Jacobian spectral norm is softly penalized during training on early times (e.g. $[0.1, 0.3]$). We fix as setup loss $\mathcal{L}_{FM}$, parametrization $\mathcal{C}_{I+NN}$ and additional regularization denoted $\mathcal{R}_{[0.1, 0.3]}$. Interestingly, this model matches (almost) the best performing model (standard FM) in FID with a Jacobian spectral norm that is twice the difference in interval $[0.1, 0.3]$ (see Figures 19 and 20). Conducting the mirror experiment, with regular-

ization applied at intermediate times, e.g. $\mathcal{R}_{[0.4\text{-}0.8]}$ applied on the interval $[0.4, 0.8]$, leads to degraded generation performance. More precisely, while regularizing at early times produces a model with similar FID than the FM baseline but that can generate visually different samples (see Figure 23), applying regularization at intermediate times instead produces samples closer to the baseline models but with degraded FID, confirming the behaviors observed under artificial perturbations. We push further these experiments on regularized models in Appendix H.1.

In the same spirit of dissecting which temporal regions of the trajectory matter most, we now train models with *ad-hoc* loss weightings that deliberately bias learning toward specific times. We build a new denoiser with weighting $w_t^{\text{mid}} = \frac{1}{(0.5-t)^2}$, putting the emphasis on accurate denoising at $t = 0.5$ whereas traditional weights focus on small and large $t$ (Kim et al., 2025, Fig. 2). We test and analyze other *mid weightings* in Appendix H.2.

Figure 8, displaying the distance between generated samples starting from the same noise, shows that $w_t^{\text{mid}}$ induces substantial deviations from both FM/den models, while generating points that are roughly at the same distance of the dataset, and having an FID similar to $\mathcal{L}_{\text{den}}$ (full evaluation is in Appendix D). This model strongly differs from the others (see also generated samples in Figure 24, Figure 25), an interesting fact as several works suggest, on the contrary, that all models, irrespective of architecture and optimization (Niedoba et al., 2025; Zhang et al., 2023), and even subset of training data (Kadkhodaie et al., 2024) *end up generating the same data.*

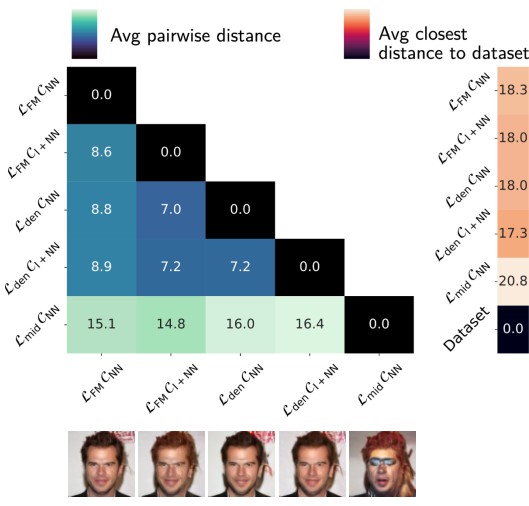

Figure 8: Left: average pairwise distance inter-models computed on 500 samples, sharing same $x_0$ across models (CelebA-64). Right: average distance of samples to train set. $\mathcal{L}_{\text{mid}}$ produces samples that differ from those of other models.

It follows from this study that:

1. By acting on the early/mid time of the generation process, we are able to change the samples generated while maintaining reasonably low FIDs. Our general denoiser framework makes it possible to explicitly build such models.

2. We show a gap between the closed-form temporal properties (early ODE trajectories splitting vs. denoising with final image already determined) and the behavior of learned generative models, where the intermediate regime matters more.

## 7    CONCLUSION

While flow matching can in principle be equivalently recast as a denoising task, we showed that connecting them also reveals how alternative choices lead to substantial variations in model behaviour. Overall, our experiments reveal that the relationship between denoising accuracy and generative performance is more subtle and complex than it may appear: it is possible to construct denoisers with degraded denoising performance without affecting the FID. Our analysis shows that engineered perturbations affect models differently depending on when they occur: drift-type perturbations are most impactful early, while noise-type ones mostly impact later stages of the process. Comparing regularity dynamics of target and learned velocities further confirms this temporal asymmetry and shows the importance of intermediate times for generation with learned models. Incidentally, we observe that models with similar FID scores can nonetheless display distinct generative behaviours. Looking ahead, the importance of parametrization choices ($\mathcal{C}_{\text{NN}}$ vs. $\mathcal{C}_{\text{I+NN}}$), emphasized in our results, deserves further exploration – for instance through gradient-step denoisers. To facilitate such investigations, we will release our code and complete toolbox of trained models, providing the community with a resource to probe generative models beyond standard benchmarks.

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

# A    RELATED WORKS

Contrary to most previous works, we do not seek to further refine the loss function or propose new parameterizations. Instead, we take the opposite approach: rather than hypothesizing which timesteps are most important and designing new losses accordingly, we fix several existing weighting schemes and systematically analyze their impact on generation behavior. Our goal is twofold: (i) to determine whether the generation process can be interpreted as a form of classical denoising operating across a broader range of noise levels, and (ii) to dissect the distinct temporal phases that occur during generation.

**Weighting strategies in denoising score matching.**    We now discuss related works within the diffusion framework, which is known to be equivalent to flow matching when the source distribution $p_0$ is Gaussian (Gao et al., 2025). For simplicity, we consider the variance-preserving diffusion process where $x_t$ is obtained from a clean sample $x_0$ and a Gaussian noise $\varepsilon \sim \mathcal{N}(0, I_d)$ as $x_t := \alpha_t x_0 + \sigma_t \varepsilon$ with $\alpha_t^2 + \sigma_t^2 = 1$. Note that we use the standard diffusion notations in this section only, meaning that the sampling process evolves from $t = T$ (noise) to $t = 0$ (clean image), as opposed to the flow matching convention where $t$ evolves from 0 to 1. Regarding parametrization, most diffusion implementations train a network to predict the noise $\varepsilon$. Some papers (Salimans & Ho, 2022; Hang et al., 2023) also study $v$-prediction (i.e. $\mathcal{C}_{\text{I+NN}}$) or $x$-prediction (i.e. $\mathcal{C}_{\text{NN}}$).

Regarding weighting, most diffusion papers follow the formulation from Ho et al. (2020) which uses an unweighted loss in the $\varepsilon$-prediction, i.e. $\mathbb{E}_{t,x_0,\varepsilon}[\|\varepsilon - \varepsilon^\theta(x_t, t)\|^2]$, corresponding in the denoising framework to the weighting $\alpha_t^2/\sigma_t^2$, equal to the signal-to-noise ratio. A line of work is devoted to *crafting* loss weightings. Salimans & Ho (2022) propose the weighting $\max(\frac{\alpha_t^2}{\sigma_t^2}, 1)$ while Yu et al. (2024) use weighting $\frac{\alpha_t}{\sigma_t}$. Both approaches assign greater importance to large noise levels, arguing that these correspond to more difficult denoising tasks and play a crucial role in error propagation during sampling. Interpreting diffusion training as multitask optimization, Hang et al. (2023) observe conflict between different timesteps objectives and suggest weighting $\min(\frac{\alpha_t^2}{\sigma_t^2}, \gamma)$ to avoid putting too much weight on the low noise levels, identified as an easy denoising task. In contrast, Choi et al. (2022) introduce the P2 weighting that puts more weight on the intermediate times, hypothesizing that perceptual features emerge during this *content phase*, as opposed to the early *coarse phase* or the late *cleanup phase*, where little noise remains in the image.

Rather than hypothesizing which timesteps are important or directly relating the weighting strategy to FID performance, we introduce an intermediate test metric–the PSNR evaluated at each timestep–which enables a more fine-grained analysis of how different timesteps contribute to generation quality. Similar to Choi et al. (2022), we find that effective denoising at intermediate times is crucial for high-quality generation.

**Unifying perspectives on loss weightings.**    Kingma & Gao (2023) provide a unifying view on a wide range of loss weightings (including FM and the previous mentioned ones) by rewriting them as differently weighted ELBO formulations. Kumar et al. (2025) also lay down the different losses (noise prediction, score prediction, etc) and the loss weighting they induce.

Nevertheless, they works do not analyze *why* different weighting choices can affect FID performance. We address this question through our denoising toolkit.

**Linking diffusion and denoising.**    Beyond standard diffusion, Delbracio & Milanfar (2023) study a broader "degradation-to-clean" setting (which is not limited to Gaussian denoising) and pick a standard denoising loss (i.e. weighting $w_t^{\text{den}} = 1$, parametrization $\mathcal{C}_{\text{NN}}$). For diffusion, they report good results on CelebA ($64 \times 64$), yet below state-of-the-art levels, which is consistent with our observations. Leclaire et al. (2025) also provide a synthetic overview of the connections between diffusion models and classical additive Gaussian denoising, interpreting diffusion as an iterative "Noising-Relaxed Denoising" process to better understand the role of noise schedules.

**Analysis of the generation phases.**    Several works study the generation process to shed light on generalisation, i.e. why they are able to generate samples that do not belong to the training dataset. This question is closely linked to how trained models approximate the exact minimizer of the training loss, which admits a closed-form expression and which can only reproduce training samples.

One line of research focuses directly on the closed-form, either theoretically (Biroli et al., 2024) or empirically (Bertrand et al., 2025). Both works highlight a critical time, at early timesteps, beyond which the closed-form points towards a single training example, while the trained models begin to deviate from it.

Another group of works seeks to understand the reasons for generalisation by characterizing the velocities or scores that are actually learned. Kadkhodaie et al. (2024); Niedoba et al. (2025); Kamb & Ganguli (2025) analyze the effective receptive field of trained models and show that it evolves from global (at high noise levels) to local (at low noise levels). Niedoba et al. (2025); Kamb & Ganguli (2025) both argue that what is actually learned by the models is a patch-wise version of the closed-form, with patch size determined by the model's receptive field: they demonstrate that this new formulation can predict in certain cases, without training, the samples learned by the model.

Beyond works focusing on the closed-form and its approximation, Sclocchi et al. (2025) connect the diffusion generative process to the learning of high- and low-level features: they specifically show that there exists a critical time at which image class is determined, while low-level features evolve smoothly throughout the generation process.

Finally, some works explain generalisation by imperfect or early-stopped optimization (Wu et al., 2025; Bonnaire et al., 2025; Favero et al., 2025).

In contrast to previous analyses that study the generation phases by measuring the deviation from the closed-form solution (based on the finite training data), we adopt a test-time denoising perspective. Rather than relying on a purely training-oriented metric, we evaluate the PSNR on test data to indirectly measure the deviation from the ideal MMSE denoiser $\mathbb{E}_{x_1 \sim p_1}[x_1 \mid x_t, t]$. This approach provides a fine-grained, time-specific analysis of how well the model performs denoising at each timestep, and how this relates to overall generative quality (e.g., FID).

## B  DETAILS ON THE LOSSES

**Flow matching denoising loss**  Substituting the velocity $v(x, t) = \frac{D(x,t) - x}{1-t}$ into the standard Flow Matching loss (Eq. (2)) together with $x_1 - x_0 = \frac{x_1 - x_t}{1-t}$ yields

$$
\begin{aligned}
\mathbb{E}_{\substack{t \sim \mathcal{U}([0,1]) \\ x_0 \sim p_0, x_1 \sim p_1}} \left[ \|v_t(x_t) - (x_1 - x_0)\|^2 \right] &= \mathbb{E}_{\substack{t \sim \mathcal{U}([0,1]) \\ x_0 \sim p_0, x_1 \sim p_1}} \left[ \left\| \frac{D_t(x_t) - x_t}{1-t} - (x_1 - x_0) \right\|^2 \right] \\
&= \mathbb{E}_{\substack{t \sim \mathcal{U}([0,1]) \\ x_0 \sim p_0, x_1 \sim p_1}} \left[ \left\| \frac{D_t(x_t) - x_t - (x_1 - x_t)}{1-t} \right\|^2 \right] \\
&= \mathbb{E}_{\substack{t \sim \mathcal{U}([0,1]) \\ x_0 \sim p_0, x_1 \sim p_1}} \left[ \frac{1}{(1-t)^2} \|D_t(x_t) - x_1\|^2 \right] \\
&= \mathcal{L}_{\mathrm{FM}}(D)
\end{aligned}
$$

**Classical denoising loss.**  We now compare the setting of *generative denoisers* that take $x_t = (1 - t)x_0 + tx_1$ as input with that of *classical denoisers* that take

$$
x_\sigma = x_1 + \sigma x_0
$$

as input. A classical denoiser $\tilde{D}$ is usually trained by minimizing

$$
\mathcal{L}(\tilde{D}) = \mathbb{E}_{\substack{\sigma \sim \mathcal{U}([0,\sigma_{\max}]) \\ x_0 \sim p_0, x_1 \sim p_1}} \left[ \|\tilde{D}(x_\sigma, \sigma) - x_1\|^2 \right], \tag{13}
$$

where $\sigma_{\max}$ is typically around 0.5. From Remark 1, *classical* and *generative* denoisers are equivalent up to the reparameterization $t(\sigma) = \frac{1}{1+\sigma}$. A change of variables in (13) gives:

$$
\begin{aligned}
\mathcal{L}(\tilde{D}) &= \mathbb{E}_{\substack{x_0 \sim p_0 \\ x_1 \sim p_1}} \left[ \int_0^{+\infty} \|\tilde{D}(x_\sigma, \sigma) - x_1\|^2 \mathbf{1}_{[0,\sigma_{\max}]}(\sigma) \mathrm{d}\sigma \right] \\
&= \mathbb{E}_{\substack{x_0 \sim p_0 \\ x_1 \sim p_1}} \left[ \int_0^1 \|\tilde{D}(x_t/t, (1-t)/t) - x_1\|^2 \mathbf{1}_{[1/(1+\sigma_{\max}),1]}(t) \frac{1}{t^2} \mathrm{d}t \right] \\
&= \mathbb{E}_{\substack{t \sim \mathcal{U}([(1+\sigma_{\max})^{-1},1]) \\ x_0 \sim p_0, x_1 \sim p_1}} \left[ \frac{1}{t^2} \|D(x_t, t) - x_1\|^2 \right], \\
&= \mathcal{L}_{\mathrm{classic}}(D),
\end{aligned}
$$

where in the before last line we used $D(x_t, t) = \tilde{D}\left(\frac{x_t}{t}, \frac{1-t}{t}\right)$.

## C  DETAILS ON TRAINING GENERATION AND METRICS

All networks are trained with the same random initialization to ensure comparability. For CIFAR-10 we train for 1000 epochs with batch size 128, and for CelebA-64 and CelebA-128, we train for 300 epochs with batch size 128. We apply exponential moving average to stabilize training. For CelebA-64 and CelebA-128, we use a U-Net architecture (Ronneberger et al., 2015) as in Ho et al. (2020). For CIFAR-10, we adopt the architecture from the `torchcfm` library (Tong et al., 2024).

The 10 denoisers of Figure 2 are trained independently with 100 epochs each. All samples are generated by solving the ODE using the dopri5 scheme from $t = 0$ to $t = 1$ with 100 timesteps; generating samples using a denoiser $D$ means that using the velocity $v(x,t) = \frac{D(x,t)-x}{1-t}$.

To measure generation quality, we use the standard Fréchet Inception Distance (Heusel et al., 2017) with the Inception-V3 features (Szegedy et al., 2016); as recommended by Stein et al. (2023) we also compute it with the DINOv2 embedding.

## D  ADDITIONAL RESULTS ON CIFAR10 AND CELEBA-64

Table 2: PSNR and FID for the different losses, to be compared with the standard FM (bottom line) . PSNR computed on 1000 images; FID on 50k train images; CIFAR-10, 1000 epochs.

| Loss | | Class | PSNR (↑) | | | | | FID / DINO (train 50k) (↓) |
|---|---|---|---|---|---|---|---|---|
| | | | $t=0.1$ | $t=0.3$ | $t=0.6$ | $t=0.9$ | $t=0.95$ | |
| $\mathcal{L}_{\text{FM}}$ | $w_t = \frac{1}{(1-t)^2}$ | $\mathcal{C}_{\text{NN}}$ | 14.39 | 18.20 | 23.50 | 32.96 | 37.41 | 4.89 / 293.84 |
| $\mathcal{L}_{\text{den}}$ | $w_t = 1$ | $\mathcal{C}_{\text{NN}}$ | 14.39 | 18.20 | 23.40 | 32.64 | 36.85 | 16.02 / 539.77 |
| $\mathcal{L}_{\text{classic}}$ | $w_t = \frac{1}{t^2}\mathbf{1}_{t>t_{\min}}$ | $\mathcal{C}_{\text{NN}}$ | 14.39 | 18.03 | 22.89 | 30.82 | 33.18 | 71.07 / 1378.22 |
| $\mathcal{L}_{\text{den}}$ | $w_t = 1$ | $\mathcal{C}_{\text{I+NN}}$ | 14.39 | 18.20 | 23.39 | 32.67 | 36.91 | 10.29 / 527.02 |
| $\mathcal{L}_{\text{classic}}$ | $w_t = \frac{1}{t^2}\mathbf{1}_{t>t_{\min}}$ | $\mathcal{C}_{\text{I+NN}}$ | 14.39 | 18.03 | 22.90 | 31.00 | 34.46 | 116.18 / 1500.15 |
| $\mathcal{L}_{\text{SNR}}$ | $w_t = \frac{t^2}{(1-t)^2}$ | $\mathcal{C}_{\text{I+NN}}$ | 14.37 | 18.19 | 23.54 | **33.05** | **37.53** | **3.71 / 181.59** |
| $\mathcal{L}_{\text{high-SNR}}$ | $w_t = \frac{t^2}{(1-t)^4}$ | $\mathcal{C}_{\text{I+NN}}$ | 13.99 | 17.84 | 23.24 | 32.98 | **37.53** | 36.66 / 955.25 |
| $\mathcal{L}_{\text{mid 0.2}}$ | $w_t = \frac{1}{(0.2-t)^2}$ | $\mathcal{C}_{\text{I+NN}}$ | 14.36 | **18.22** | 23.47 | 32.75 | 37.06 | 8.07 / 432.52 |
| $\mathcal{L}_{\text{mid 0.3}}$ | $w_t = \frac{1}{(0.3-t)^2}$ | $\mathcal{C}_{\text{I+NN}}$ | 14.26 | **18.22** | 23.51 | 32.81 | 37.13 | 8.17 / 361.39 |
| $\mathcal{L}_{\text{mid 0.4}}$ | $w_t = \frac{1}{(0.4-t)^2}$ | $\mathcal{C}_{\text{I+NN}}$ | 14.14 | 18.17 | 23.54 | 32.85 | 37.20 | 9.80 / 351.951 |
| $\mathcal{L}_{\text{mid 0.5}}$ | $w_t = \frac{1}{(0.5-t)^2}$ | $\mathcal{C}_{\text{I+NN}}$ | 13.93 | 18.05 | **23.56** | 32.91 | 37.27 | 9.05 / 313.65 |
| $\mathcal{L}_{\text{mid 0.6}}$ | $w_t = \frac{1}{(0.6-t)^2}$ | $\mathcal{C}_{\text{I+NN}}$ | 14.00 | 17.93 | **23.56** | 32.96 | 37.33 | 11.25 / 339.08 |
| $\mathcal{L}_{\text{mid 0.7}}$ | $w_t = \frac{1}{(0.7-t)^2}$ | $\mathcal{C}_{\text{I+NN}}$ | 13.96 | 17.81 | 23.51 | 33.01 | 37.40 | 19.39 / 457.23 |
| $\mathcal{L}_{\text{mid 0.8}}$ | $w_t = \frac{1}{(0.8-t)^2}$ | $\mathcal{C}_{\text{I+NN}}$ | 14.01 | 17.74 | 23.36 | 33.04 | 37.46 | 27.78 / 632.75 |
| $\mathcal{L}_{\text{mid 0.9}}$ | $w_t = \frac{1}{(0.5-t)^2}$ | $\mathcal{C}_{\text{I+NN}}$ | 14.27 | 17.80 | 23.09 | 33.00 | 37.50 | 35.94 / 986.16 |
| $\mathcal{L}_{\text{mid 0.95}}$ | $w_t = \frac{1}{(0.5-t)^2}$ | $\mathcal{C}_{\text{I+NN}}$ | 14.33 | 17.93 | 23.08 | 32.88 | 37.48 | 37.82 / 1097.20 |
| $\mathcal{L}_{\text{FM}} + \mathcal{R}_{[0.1,0.3]}$ | $w_t = \frac{1}{(1-t)^2}$ | $\mathcal{C}_{\text{I+NN}}$ | 14.32 | 18.19 | 23.54 | 33.01 | 37.47 | 6.47 / 279.18 |
| $\mathcal{L}_{\text{FM}} + \mathcal{R}_{[0.3,0.6]}$ | $w_t = \frac{1}{(1-t)^2}$ | $\mathcal{C}_{\text{I+NN}}$ | 14.36 | 17.96 | 23.41 | 33.01 | 37.46 | 24.18 / 601.15 |
| $\mathcal{L}_{\text{FM}} + \mathcal{R}_{[0.6,0.9]}$ | $w_t = \frac{1}{(1-t)^2}$ | $\mathcal{C}_{\text{I+NN}}$ | 14.38 | 18.20 | 23.31 | 32.86 | 37.41 | 29.58 / 564.66 |
| $\mathcal{L}_{\text{FM}} + \mathcal{R}_{[0.4,0.8]}$ | $w_t = \frac{1}{(1-t)^2}$ | $\mathcal{C}_{\text{I+NN}}$ | 14.38 | 18.18 | 23.47 | 33.01 | 37.47 | 16.04 / 459.39 |
| 10-denoisers $\mathcal{L}_{\text{den}}$ | $w_t = 1$ | $\mathcal{C}_{\text{I+NN}}$ | 14.38 | 18.15 | 23.43 | 32.98 | 37.45 | 20.81 / 485.50 |
| $\mathcal{L}_{\text{FM}}$ | $w_t = \frac{1}{(1-t)^2}$ | $\mathcal{C}_{\text{I+NN}}$ | 14.39 | 18.21 | 23.54 | 33.02 | 37.47 | 3.90 / 210.54 |

Table 3: PSNR and FID for the different losses, to be compared with the standard FM (bottom line). PSNR computed on 1000 test images; FID on 50k train images; CelebA-64, 300 epochs.

| Loss | | Class | PSNR (↑) | | | | | FID / DINO (train 50k) (↓) |
|---|---|---|---|---|---|---|---|---|
| | | | $t = 0.1$ | $t = 0.3$ | $t = 0.6$ | $t = 0.9$ | $t = 0.95$ | |
| $\mathcal{L}_{\text{den}}$ | $w_t = 1$ | $\mathcal{C}_{\text{NN}}$ | 16.01 | 20.98 | 26.25 | 34.50 | 37.91 | 19.87 / 518.03 |
| $\mathcal{L}_{\text{classic}}$ | $w_t = \frac{1}{t^2}\mathbf{1}_{t > t_{\min}}$ | $\mathcal{C}_{\text{NN}}$ | 15.88 | 20.19 | 24.41 | 29.35 | 29.81 | 85.44 / 1708.95 |
| $\mathcal{L}_{\text{mid}}$ | $w_t = \frac{1}{(0.5-t)^2}$ | $\mathcal{C}_{\text{NN}}$ | 15.29 | 20.82 | 26.54 | 35.17 | 38.95 | 20.06 / 434.22 |
| $\mathcal{L}_{\text{FM}}$ | $w_t = \frac{1}{(1-t)^2}$ | $\mathcal{C}_{\text{NN}}$ | 15.93 | 20.79 | 26.12 | 34.83 | 38.81 | 14.54 / 448.83 |
| $\mathcal{L}_{\text{den}}$ | $w_t = 1$ | $\mathcal{C}_{\text{I+NN}}$ | 15.98 | 20.87 | 26.10 | 34.40 | 38.06 | 18.83 / 513.82 |
| $\mathcal{L}_{\text{classic}}$ | $w_t = \frac{1}{t^2}\mathbf{1}_{t > t_{\min}}$ | $\mathcal{C}_{\text{I+NN}}$ | 15.89 | 20.26 | 24.70 | 31.34 | 34.67 | 133.93 / 1374.11 |
| $\mathcal{L}_{\text{mid}}$ | $w_t = \frac{1}{(0.5-t)^2}$ | $\mathcal{C}_{\text{I+NN}}$ | 14.80 | 20.79 | **26.58** | 35.22 | 39.08 | 19.37 / 412.68 |
| $\mathcal{L}_{\text{FM}}$ | $w_t = \frac{1}{(1-t)^2}$ | $\mathcal{C}_{\text{I+NN}}$ | 16.03 | 21.08 | 26.52 | **35.33** | **39.34** | 4.45 / 164.77 |

# E SENSITIVITY OF GENERATION TO LAST TIMESTEPS

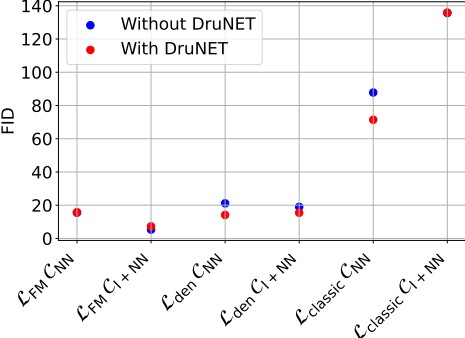

Figure 9: FID (10k-test) comparison on Celeba64 when replacing late-time denoising with a pre-trained GS-DRUNet. While the external denoiser slightly improves performance, a gap with the FM baseline remains, suggesting late-time denoising is not the main cause of the discrepancy.

Given the sensitivity of FID to noise, one might wonder whether the poor performance of some denoisers in our toolkit is due to their limited denoising ability at late times (i.e., low noise levels). The following experiment Figure 9 shows that this is not the case. We generate samples as usual with a Dopri5 scheme, but stop the integration at $t = 0.95$ and complete the process with a generic GS-DRUNet pre-trained denoiser from the deepinv library (Tachella et al., 2025), trained on a different dataset. We then compute the FID on 10k generated images and compare it with the original FID. Although plugging in this external denoiser slightly improves the FID, a gap with the FM baseline remains, indicating that late-time denoising is not the only cause of the discrepancy.

# F INPAINTING VISUAL RESULTS

We display in Figure 10 the reconstructions obtained with our denoisers on a single inpainting task on CelebA-64 with a mask of size $17 \times 17$. We set the parameter $\alpha$ in PnP-Flow to $\alpha = 0.3$ and the number of iterations to 100, following the recommendations of the original paper (Martin et al., 2025) and using their public implementation. One can see that only a few methods accurately recover the headband: namely the flow-matching baseline $\mathcal{L}_{\text{FM}}, \mathcal{C}_{\text{I+NN}}$ and the variants $\mathcal{L}_{\text{mid}}, \mathcal{C}_{\text{NN}}$ and $\mathcal{L}_{\text{mid}}, \mathcal{C}_{\text{I+NN}}$.

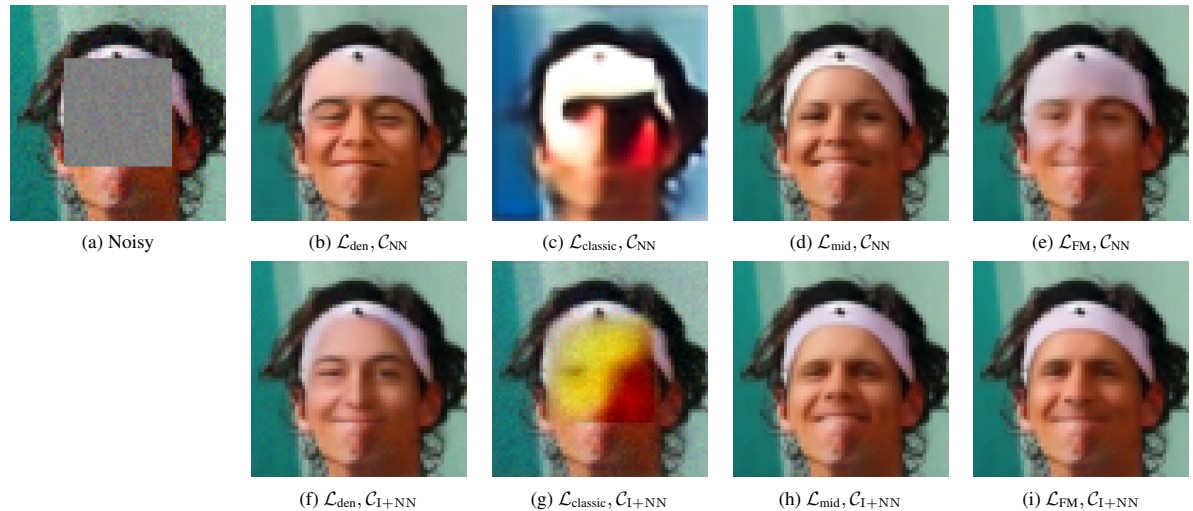

Figure 10: Inpainting results: top row $\mathcal{C}_{\text{NN}}$, bottom row $\mathcal{C}_{\text{I+NN}}$, columns correspond to losses.

## G  ABLATION ON THE PERTURBATION EXPERIMENTS SECTION 5

Despite the checkboard perturbations, we also consider below a range of perturbations defined as follows:

- Low-frequency - $r$ denotes the perturbation $\delta := h \star g_0$ where $g_0$ is a fixed random Gaussian noise and $h$ is a low-pass filter with cutoff frequency $r$;

- Low-frequency-luminance - $r$ same as above, except $g_0$ is shared across the 3 RGB channels (i.e., grayscale noise), and $h$ is a low-pass filter with cutoff frequency $r$.

The higher $r$ is, the more frequency are present (hence evolves from low-freq at $r = 0.05$ to high-freq at $r = 0.5$) Results on CelebA-128 are in Figure 11, samples are displayed Figures 14 and 15. Results on CIFAR10 are in Figures 16 and 17, samples are displayed Figure 18.

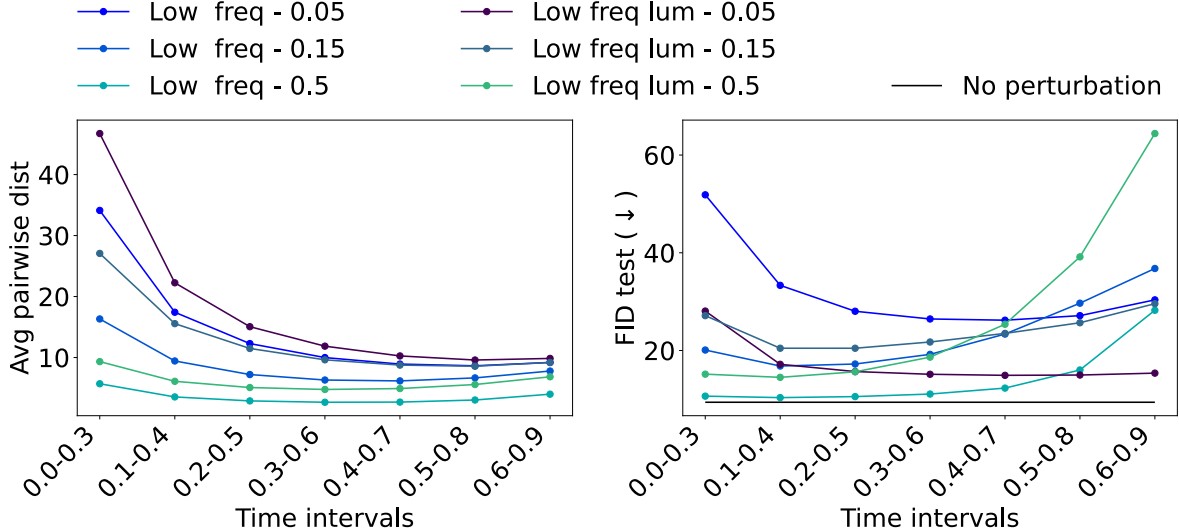

Figure 11: Influence of different perturbations at different generation phases on the FID (10K, test) on CelebA 128.

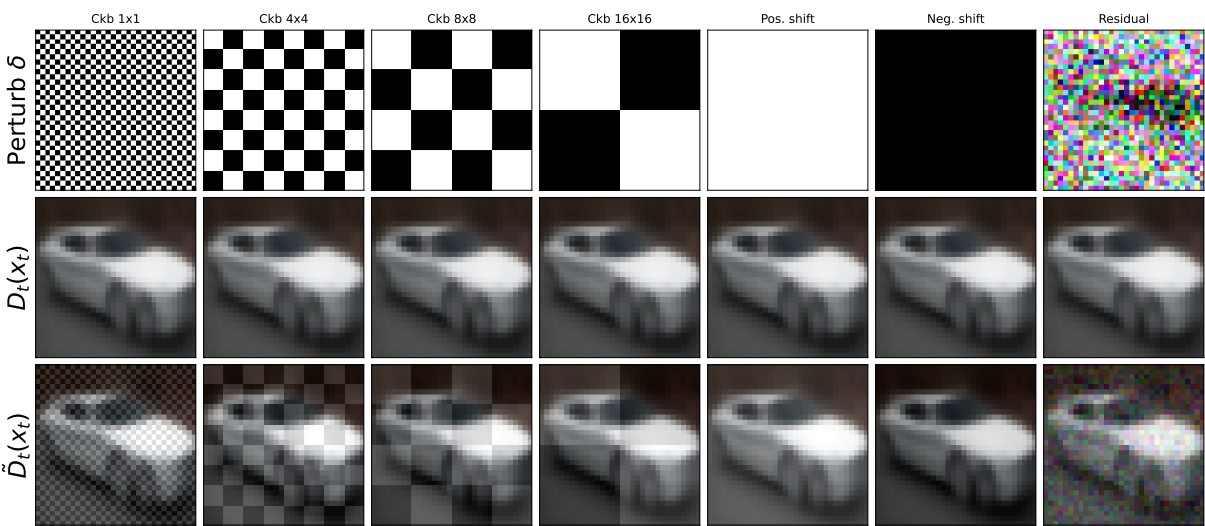

Figure 12: Perturbations applied to denoiser $D_t$ (here for $t = 0.3$). Experiments done on CIFAR-10.

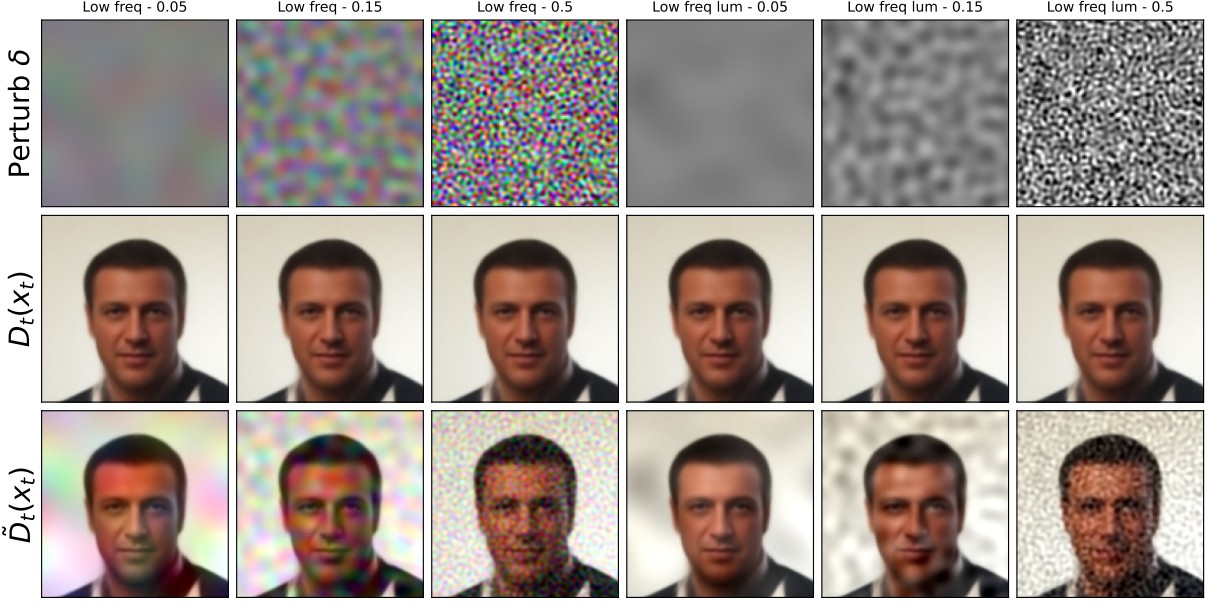

Figure 13: Frequency-based perturbations applied to denoiser $D_t$ (here for $t = 0.3$). Experiments done here on CelebA 128.

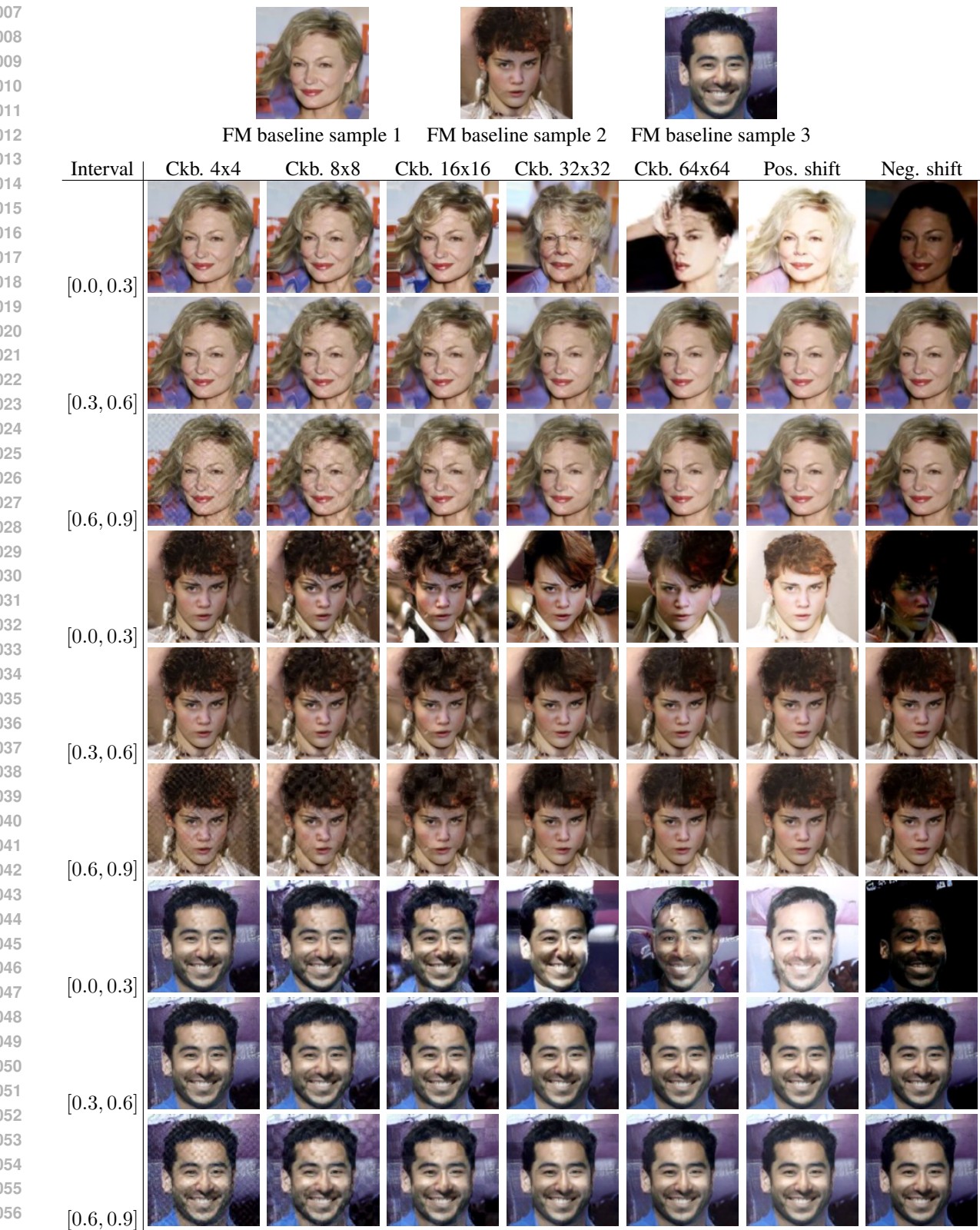

Figure 14: Effect of perturbations on generated CelebA $128 \times 128$ samples.

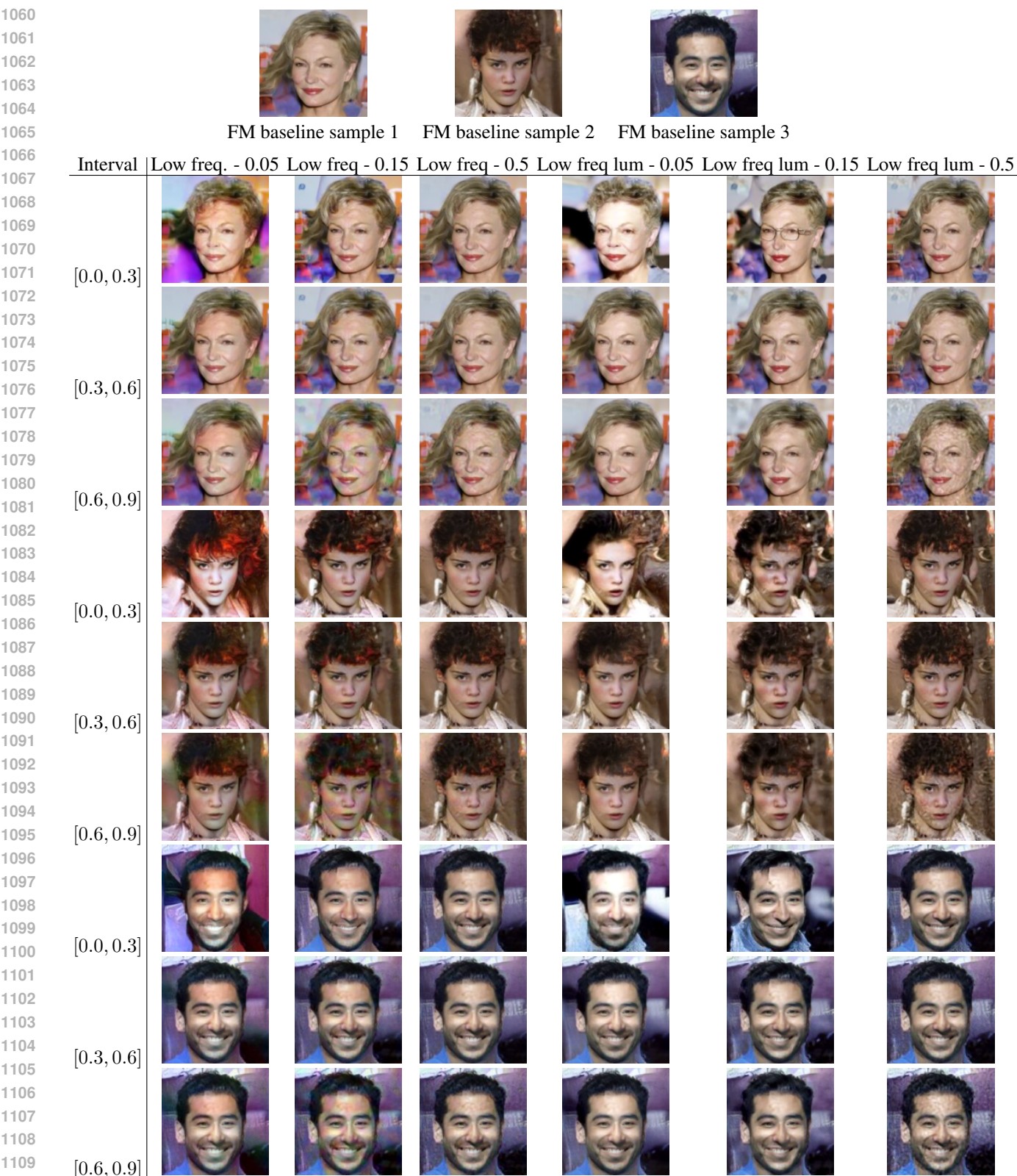

Figure 15: Effect of frequency-domain perturbations on CelebA $128 \times 128$ samples.

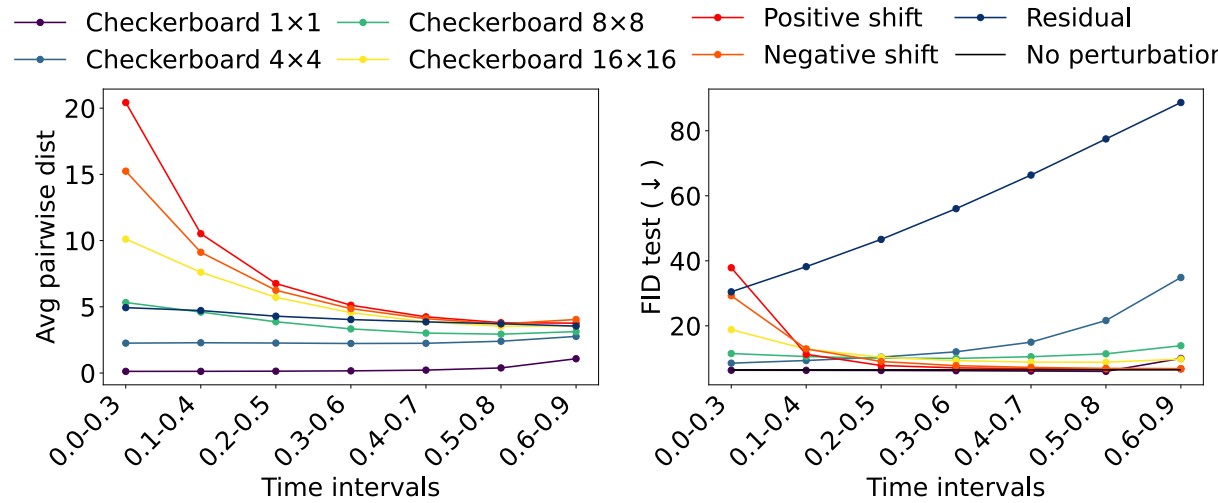

Figure 16: Influence of different perturbations at different generation phases on the FID (10K, test) on CIFAR10.

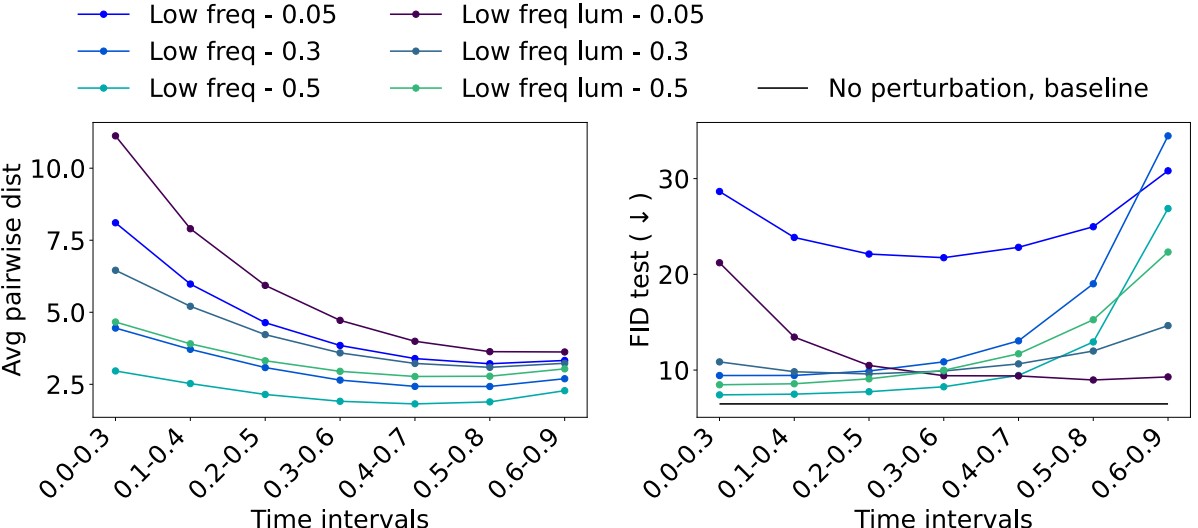

Figure 17: Influence of different perturbations at different generation phases on the FID (10K, test) on CIFAR10.

# H  ADDITIONAL EXPERIMENTS ON GENERATION PHASES

## H.1  TIME-INTERVAL JACOBIAN PENALIZED MODELS

**Experimental setup   Our goal here is to induce a controlled decrease of the Jacobian norm over selected time intervals and study how this affects denoising and generation.**   We train standard flow matching models ($\mathcal{L}_{\text{FM}}, \mathcal{C}_{\text{I+NN}}$) with additional Jacobian spectral norm regularization applied over a prescribed time interval $[t_{\min}, t_{\max}]$:

$$\mathcal{R}_{[t_{\min}, t_{\max}]}(\theta) := \lambda \mathbf{1}_{t \in [t_{\min}, t_{\max}]} \max \left( \|\nabla_x v_t^\theta(x_t)\|_2, M \right), \tag{14}$$

where $\lambda$ is a regularization parameter and $M$ is the targeted upper bound on the Jacobian spectral norm. The models are trained without regularization for 990 epochs and finetuned with regularization for the last 10 epochs. The Jacobian spectral norm is estimated using the power method (10 iterations). We choose the early phase interval $[0.1, 0.3]$ (see Figure 7a) and three intermediate phase intervals, $[0.3, 0.6]$, $[0.6, 0.9]$ and $[0.4, 0.8]$. The threshold

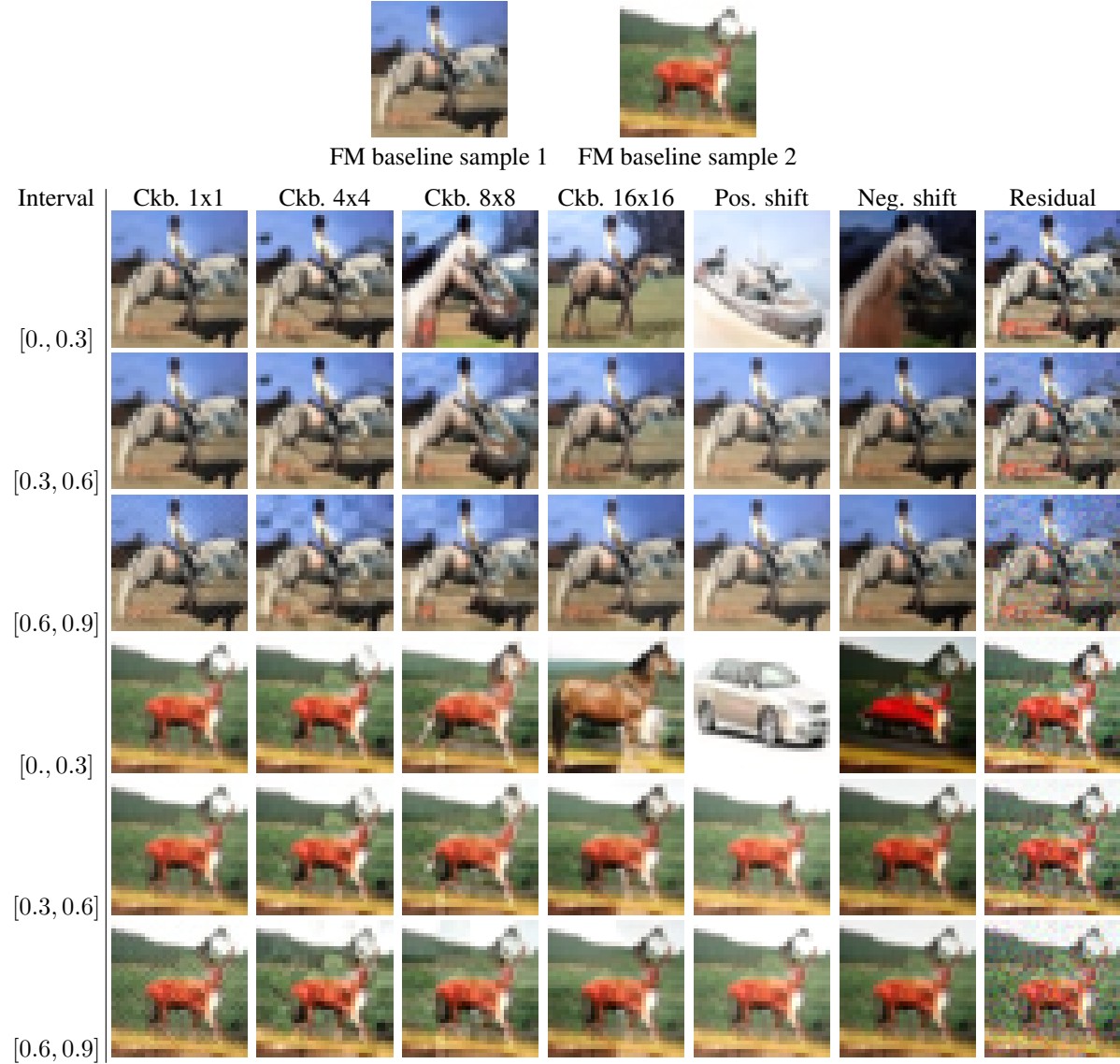

Figure 18: Effect of perturbations on generated samples (CIFAR-10).

$M$ is set to $M = 4$ for $[0.1, 0.3]$, and $M = 2$ for $[0.4, 0.8]$ and $[0.3, 0.6]$, and $M = 1$ for the $[0.6, 0.9]$. The regularization parameter is set to $\lambda = 0.1$ for $[0.1, 0.3]$, $\lambda = 0.2$ for $[0.3, 0.6]$ and $[0.4, 0.8]$, $\lambda = 0.4$ for $[0.6, 0.9]$.

**Comparison with the closed-form optimal velocity field.** The early-time regularization (i.e. $[0.1, 0.3]$) produces the largest deviation from the closed-form behaviour in terms of the mean Jacobian spectral norm measured over the ODE trajectories (see Figure 19a): the regularized model don't reproduce the sharp peak of the closed-form Jacobian around $t \approx 0.2$ and instead maintain higher values in mid times. In contrast, models regularized in the intermediate time phase ($[0.3, 0.6]$, $[0.6, 0.9]$ and $[0.4, 0.8]$) exhibit a decay of the Lipschitz constant around time $0.2 - 0.3$ getting closer to the closed-form behaviour.

**Comparison with standard Flow Matching.** In terms of **denoising performance** (Figure 20a), a decrease in the Jacobian norm over a given time interval (compared to the standard FM model) corresponds to a decrease in PSNR at the time steps within that interval. In other words, *constraining the local Lipschitz constant degrades the denoising accuracy at those times.*

In terms of **generation performance** (Figure 20b), two distinct behaviors emerge. Penalizing the Jacobian norm during the early phase yields FID scores comparable to, and slightly better than, the standard FM model. Qualitatively (see Figure 23), this produces samples that differ noticeably from the FM baseline, sometimes even altering the image class. In contrast, penalizing the model during the intermediate phase leads to significantly higher FIDs and noisier samples compared to the standard FM outputs. This is consistent with our controlled perturbation experiments in Section 5, where perturbations applied at early times tend to shift the global image distribution while maintaining a low FID, whereas perturbations applied at later times produce noisier samples and result in a more severe degradation of the FID.

To further quantify the impact of these regularizations on the generated samples, we compute the average pairwise distance between samples generated from the same $x_0$, thereby directly comparing their ODE trajectories (Figure 22a). Among all models, the largest distances (relative to the standard FM baseline) are observed for models regularized during the early phase or at the beginning of the intermediate phase.

**Discussion** Overall, this experiment indicates that shifting the peak of the Jacobian spectral norm as done with the early-time penalizations can change the ODE trajectories without altering the generation quality. In contrast, maintaining a relatively high Jacobian norm appears important in order to keep good denoising quality in the mid times, which in turn is crucial to achieve good generation quality. Notably, matching the behaviour of the closed-form velocity field in terms of the Jacobian spectral norm seems undesirable in the early/mid-time regimes: not reproducing its peak at early times still yields models with competitive FID scores, whereas getting closer to the closed-form behaviour at mid times results in degraded FIDs.

### H.2 MODELS WITH INTERMEDIATE WEIGHTING

**Experimental setup.** To dissect which temporal regions of the diffusion trajectory contribute most to the learned denoising dynamics, we extend our denoiser toolkit with a family of *ad-hoc* objectives that deliberately bias learning toward specific times. In particular, we introduce a weighting function

$$w_t^{\mathrm{mid}-t^*} = \frac{1}{(t^*-t)^2}, \tag{15}$$

which emphasizes accurate denoising around a chosen time $t^* \in [0,1]$. This formulation generalizes the intermediate weighting $w_t^{\mathrm{mid}}$ presented in Section 6.2, allowing us to probe the model's sensitivity to localized temporal emphasis along the diffusion process. As with the baseline models, we evaluate for each $w_t^{\mathrm{mid}-t^*}$ both the per-time PSNR curves (Figure 21a) and the final FID scores (Figure 21), complemented by pairwise model distance maps (Figure 22b) and the evolution of spatial regularity measured through the Jacobian spectral norm (Figure 19b).

**Observations and discussion.** Surprisingly, despite the formal resemblance between the standard flow-matching weighting $w_t^{\mathrm{FM}} = \frac{1}{1-t}^2$ and the limiting behavior of $w_t^{\mathrm{mid}-t^*}$ as $t^* \to 1$, the results diverge markedly. Instead of recovering the performance of the FM baseline, we observe a sharp degradation: the FID explodes for $t^* = 0.95$, and the PSNR deteriorates across all times. This finding indicates that it is not straightforward to isolate the influence of a single time point: poor denoising performance at other times propagates through the training dynamics, affecting even the regions of emphasis. Interestingly, the model most similar to the standard FM baseline is obtained for $t^* = 0.2$.

Two distinct degradation regimes emerge in the FID/PSNR trends, consistent with the observations of Section 5. In the first regime, where the emphasis is placed on intermediate times $t^\star \in [0.2, 0.5]$, the FID remains nearly unchanged and close to the SOTA baseline. Pairwise distances (Figure 22b) and PSNR curves (Figure 21a) show that, while denoising performance in the early phase worsens as $t^\star$ increases, this mainly induces sample drift rather than a loss in overall generation quality (see samples in Figure 24). These models appear to have learned qualitatively different denoising functions, highlighting how weighting choices can modulate trajectory alignment.

In the second regime, for larger times $t^\star \in [0.6, 0.9]$, image quality degrades sharply: both FID and intermediate-time PSNR deteriorate, and pairwise distances decrease again for $t^\star > 0.8$.

These observations suggest that while poor denoising performance in the early phase (e.g. weighting with $t^\star = 0.5$) primarily induces sample drifts, poor denoising performance in the the intermediate phase (e.g. weighting with $t^\star = 0.9$) mainly affect the generation quality.

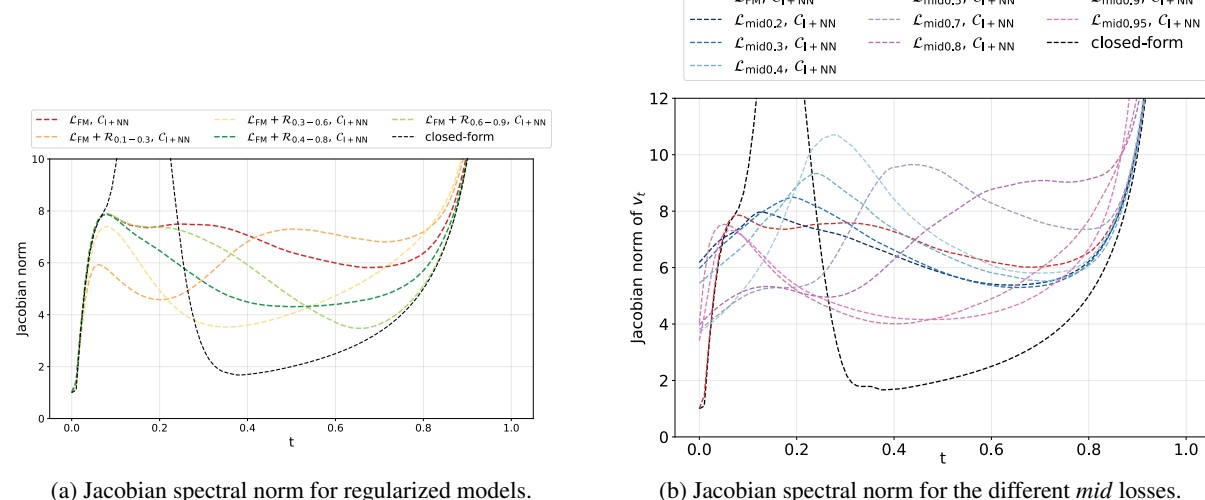

(a) Jacobian spectral norm for regularized models.

(b) Jacobian spectral norm for the different *mid* losses.

Figure 19: Mean and standard deviation of the spectral norm $\|\nabla_x v(x_t, t)\|_2$, computed on 1000 ODE trajectories $(x_t)$. CIFAR-10, 1000 epochs. The Jacobian spectral norm is estimated using the power iteration with maximum number of iteration set to 10.

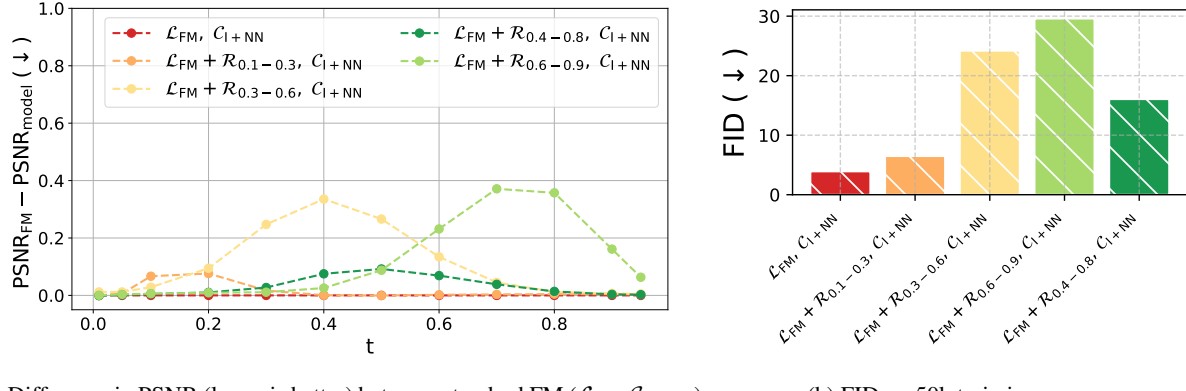

(a) Difference in PSNR (lower is better) between standard FM ($\mathcal{L}_{\mathrm{FM}}, \mathcal{C}_{\mathrm{I+NN}}$) and various models, computed on 1000 test images. Positive values indicate worse denoising performance compared to standard FM.

(b) FID on 50k train images.

Figure 20: PSNR and FID for the different *regularizations*, CIFAR-10, 1000 epochs. PSNR degradation at early times does not impair generation performance, while PSNR degradation at mid times systematically correlates with a higher FIDs.

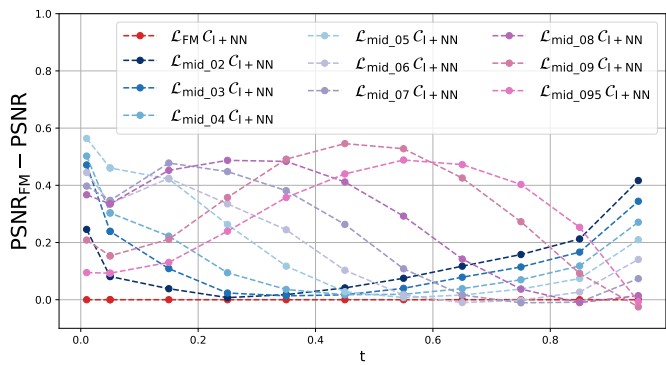
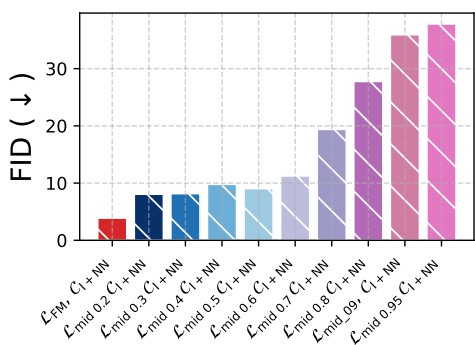

(a) Difference in PSNR (lower is better) between standard FM ($\mathcal{L}_{\text{FM}}$, $\mathcal{C}_{\text{I+NN}}$) and various models, computed on 1000 test images. Positive values indicate worse denoising performance compared to standard FM.

(b) FID on 50k train images.

Figure 21: PSNR and FID for the different *mid* losses, CIFAR-10, 1000 epochs.

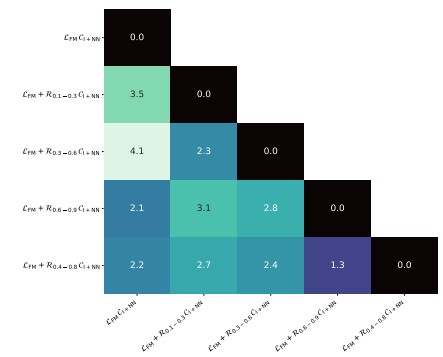
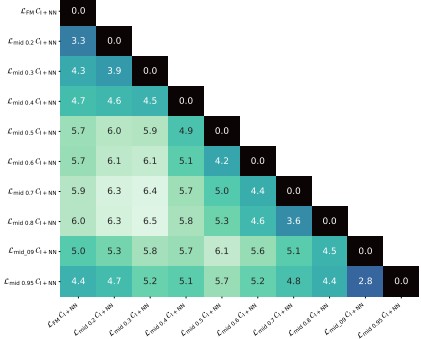

(a) Average pairwise distance for the different *Jacobian regularizations*.

(b) Average pairwise distance for the different *mid* losses.

Figure 22: Average pairwise distance inter-models computed on 1000 samples, sharing same $x_0$ across models. Experiments done on CIFAR-10, models trained with 1000 epochs.

# I  ABLATION ON THE ODE SOLVER

Here, we perform a sanity check to ensure that the choice of ODE solver does not affect the relative performance of the models. In Table 4, we compare the FID scores of six models when samples are generated using dopri5 or euler100. The results show that the ranking of the models is consistent across both solvers, indicating that our choice of dopri5 does not bias the comparison.

|  |  | Euler (100 steps) | Dorpi5 |
|---|---|---|---|
| $\mathcal{L}_{\text{FM}}$ | $\mathcal{C}_{\text{NN}}$ | 5.93 (2) | 4.89 (2) |
| $\mathcal{L}_{\text{FM}}$ | $\mathcal{C}_{\text{I+NN}}$ | 4.84 (1) | 3.90 (1) |
| $\mathcal{L}_{\text{den}}$ | $\mathcal{C}_{\text{NN}}$ | 13.37 (4) | 16.02 (4) |
| $\mathcal{L}_{\text{den}}$ | $\mathcal{C}_{\text{I+NN}}$ | 10.51 (3) | 10.29 (3) |
| $\mathcal{L}_{\text{classic}}$ | $\mathcal{C}_{\text{NN}}$ | 54.09 (5) | 71.07 (5) |
| $\mathcal{L}_{\text{classic}}$ | $\mathcal{C}_{\text{I+NN}}$ | 113.92 (6) | 116.18 (6) |

Table 4: FID 50k on train set. The numbers in blue are the ranking of the models.

## J  JUSTIFICATION OF THE FM WEIGHTING

In the following, we attempt to provide a statistical interpretation for why the standard Flow Matching weighting

$$w(t) = \frac{1}{(1-t)^2}$$

often yields the best empirical performance. This is not meant to be a formal proof, but rather a conceptual link with classical results in the regression literature (Shalizi, 2013; Schick, 1997; Aitken, 1936).

**Maximum-likelihood justification on a toy model.**   We consider the following toy generative model. Let

$$X_0 \sim \mathcal{N}(0, I), \qquad X_1 \sim \mathcal{N}(0, \tau^2 I), \qquad X_0 \perp X_1,$$

and for $t \in (0,1)$ define $X_t = (1-t)X_0 + tX_1$. Introducing $Y = X_t/t$ yields

$$Y = X_1 + \sigma(t)X_0, \qquad \sigma(t) = \frac{1-t}{t},$$

so that

$$Y \mid (X_1, t) \sim \mathcal{N}(X_1, \sigma^2(t)I).$$

In this linear–Gaussian setting, the posterior distribution admits a closed form:

$$X_1 \mid (Y = y, t) \sim \mathcal{N}\big(\mu_{\text{post}}(y,t), \, \sigma_{\text{post}}(t)I\big),$$

with

$$\sigma_{\text{post}}(t) = \left(\frac{1}{\tau^2} + \frac{1}{\sigma^2(t)}\right)^{-1}, \qquad \mu_{\text{post}}(y,t) = \frac{\tau^2}{\tau^2 + \sigma^2(t)}\, y.$$

Since our goal is to approximate the conditional expectation $\mathbb{E}[X_1 \mid X_t = x_t, t]$ (or equivalently $\mathbb{E}[X_1 \mid Y = y, t]$), we introduce a parametric function $f_\theta$ intended to model this conditional mean. We then *define* a conditional Gaussian model

$$p_\theta(x_1 \mid y, t) = \mathcal{N}\big(x_1; f_\theta(y,t), \sigma_{\text{post}}^2(t)I\big).$$

Of course, in this toy example we already know that $\mathbb{E}[X_1 \mid Y = y, t] = \mu_{\text{post}}$ but we still want to estimate it for the sake of understanding the weightings. For a single sample the negative log-likelihood (NLL) is

$$-\log p_\theta(x_1 \mid y, t) = \frac{1}{2\sigma_{\text{post}}^2(t)}\|x_1 - f_\theta(y,t)\|^2 + \frac{d}{2}\log\big(2\pi\sigma_{\text{post}}^2(t)\big),$$

so the expected NLL is, up to an additive constant,

$$\mathcal{L}_{\text{NLL}}(\theta) = \mathbb{E}\Big[\frac{1}{\sigma_{\text{post}}^2(t)}\|X_1 - f_\theta(Y,t)\|^2\Big].$$

Hence a time-dependent weighting appears naturally:

$$w(t) \propto \frac{1}{\sigma_{\text{post}}^2(t)} = \frac{1}{\tau^2} + \frac{t^2}{(1-t)^2}.$$

This weight behaves as $1/(1-t)^2$ for large $t$. The toy model therefore provides a principled statistical motivation: inverse-variance weighting based on the forward corruption noise $\sigma^2(t)$ is the maximum-likelihood choice in an analytically tractable setting, and this coincides with the empirically best-performing weighting $(1/(1-t))^2$.

**Remark 2.** We also experiment with an SNR-based weighting, given by $w_t = \frac{t^2}{(1-t)^2}$, which corresponds to the signal-to-noise ratio of $x_t$. We report the corresponding performance in Table 2. We find that this weighting performs similarly to the weighting $1/(1-t^2)$, and even slightly better. This is also consistent with empirical practices in diffusion models, where a similar weighting has been used (see the discussion in the related works, Appendix A).

**Remark 3.** If we do not assume $X_1$ to have a Gaussian prior, one may instead adopt a different modeling assumption and approximate the posterior $p(x_1 \mid y, t)$ by a Gaussian distribution with mean $\mathbb{E}[X_1 \mid Y = y, t]$ and covariance $C_t := \text{Cov}[X_1 \mid Y = y, t]$. Under this assumption, the same maximum-likelihood reasoning leads to the weighted quadratic loss

$$\mathbb{E}\Big[(X_1 - f_\theta(Y,t))^\top C_t^{-1} (X_1 - f_\theta(Y,t))\Big].$$

This loss is however intractable in practice since $C_t$ depends on the unknown data distribution.

Although this Gaussian approximation is not valid in general, it becomes increasingly reasonable near $t = 1$, where the conditional distribution $X_1 \mid Y = y, t$ concentrates around a single point. Moreover, we expect $C_t \to 0$ as $t \to 1$, which justifies the use of a weighting $w_t$ that diverges near $t = 1$.

**Connection with heteroscedastic regression in the linear case.** We now assume that the estimator $f$ is restricted to a linear form. Consider the regression model

$$X_1 = Y\beta + \varepsilon(t),$$

where $A$ is a linear operator and $\varepsilon$ is a zero-mean Gaussian noise satisfying

$$\mathbb{E}[\varepsilon \mid Y, t] = 0, \qquad \mathrm{Cov}[\varepsilon(t) \mid Y, t] = v(t)I.$$

The model is said to be *heteroscedastic* because the conditional covariance $v(t)$ is not constant but varies with $t$.

The Generalized least square result (Aitken, 1936; Greene, 2003) states that, **the best linear unbiased estimator** (referred to as *BLUE*) of $A$, **i.e. the one with minimum covariance** (in the sense of positive semidefinite ordering) is obtained by solving the generalized least-squares problem

$$\hat{A} = \arg\min_A \mathbb{E}\left[\frac{1}{v(t)}\left\|X_1 - AY\right\|^2\right].$$

Thus, in the linear case, the optimal weighting is proportional to the inverse conditional variance of the noise, $\frac{1}{v(t)}$. We can therefore see the case of Flow Matching as a generalization of this principle.

## K  FASTER TRAINING WITH THE RESIDUAL PARAMETRIZATION

Here we provide empirical evidence that the residual parametrization $\mathcal{C}_{\mathrm{I+NN}}$ not only achieves better final performance, but also converges faster, requiring fewer training epochs to reach a given FID than the $\mathcal{C}_{\mathrm{NN}}$ parametrization. In Figure 26, we report the FID (10k test) on CelebA $128 \times 128$ as a function of the number of training epochs.

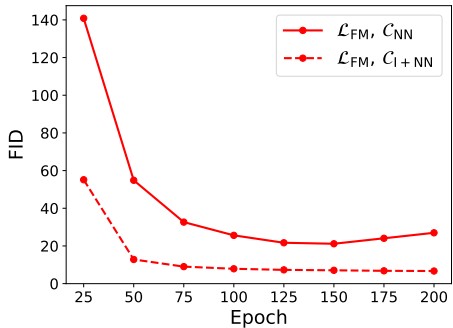

Figure 26: FID (10k test) on CelebA $128 \times 128$ as a function of training epochs.

## L  ADDITIONAL EXPERIMENT ON THE REGULARITY OF TARGET AND LEARNED MODELS

We provide in Figure 27 additional evidence of the gap between the regularity of a trained model (here, the standard flow matching baseline corresponding to loss $\mathcal{L}_{\mathrm{FM}}$ and parameterization $\mathcal{C}_{\mathrm{I+NN}}$) and the closed-form target. In this experiment, we use the TinyImagenet dataset, which is a downscaled variant of the ImageNet dataset that contains 200 classes with 500 training images per class and image resolution 64×64, using the training code and the backbone architecture provided in Lipman et al. (2024).

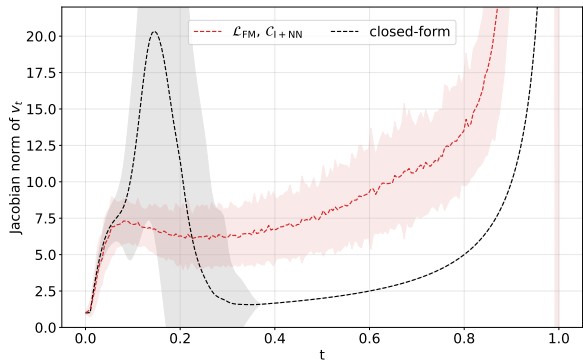

Figure 27: Mean and standard deviation of the spectral norm $\|\nabla_x v(x_t, t)\|_2$ computed on ODE trajectories (1000 points for the closed-form velocity, 400 for the standard FM trained model $\mathcal{L}_{\mathrm{FM}}, \mathcal{C}_{\mathrm{I+NN}}$) on TinyImagenet with Gaussian $p_0$.

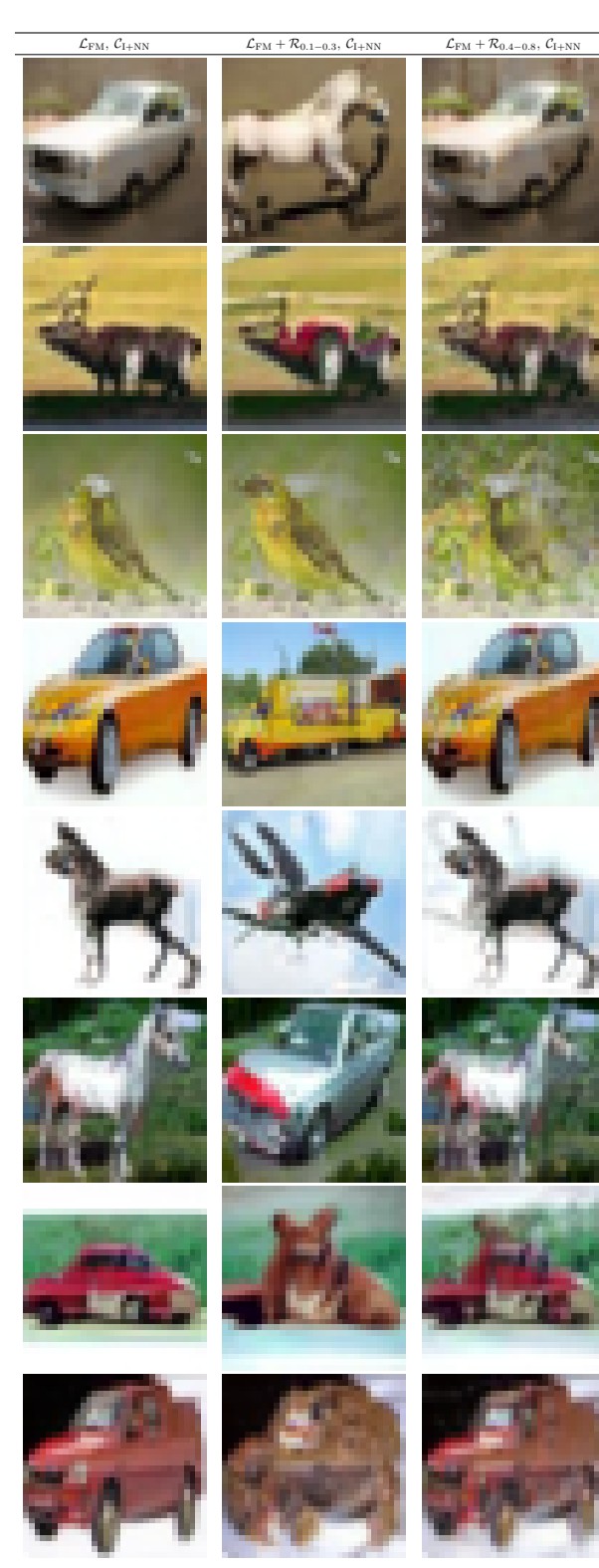

Figure 23: CIFAR-10 samples generated by models trained with different *Jacobian regularizations*. Regularizing at early times ($\mathcal{R}_{0.1-0.3}$) changes visually more the samples than applying regularization at intermediate times ($\mathcal{R}_{0.4-0.8}$).

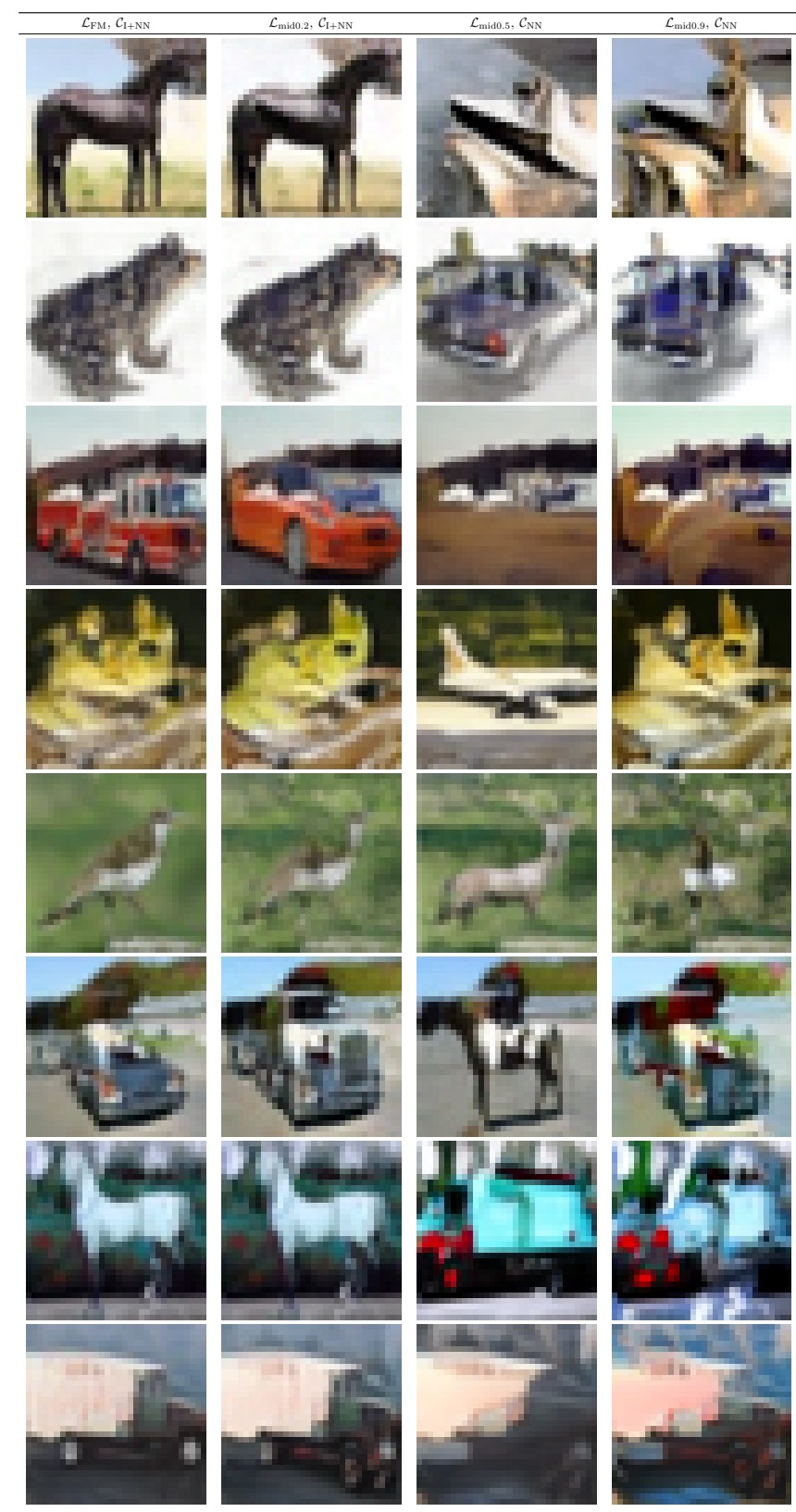

Figure 24: CIFAR-10 samples generated by models trained with different *mid weightings*. Each row corresponds to one fixed initial point $x_0$.

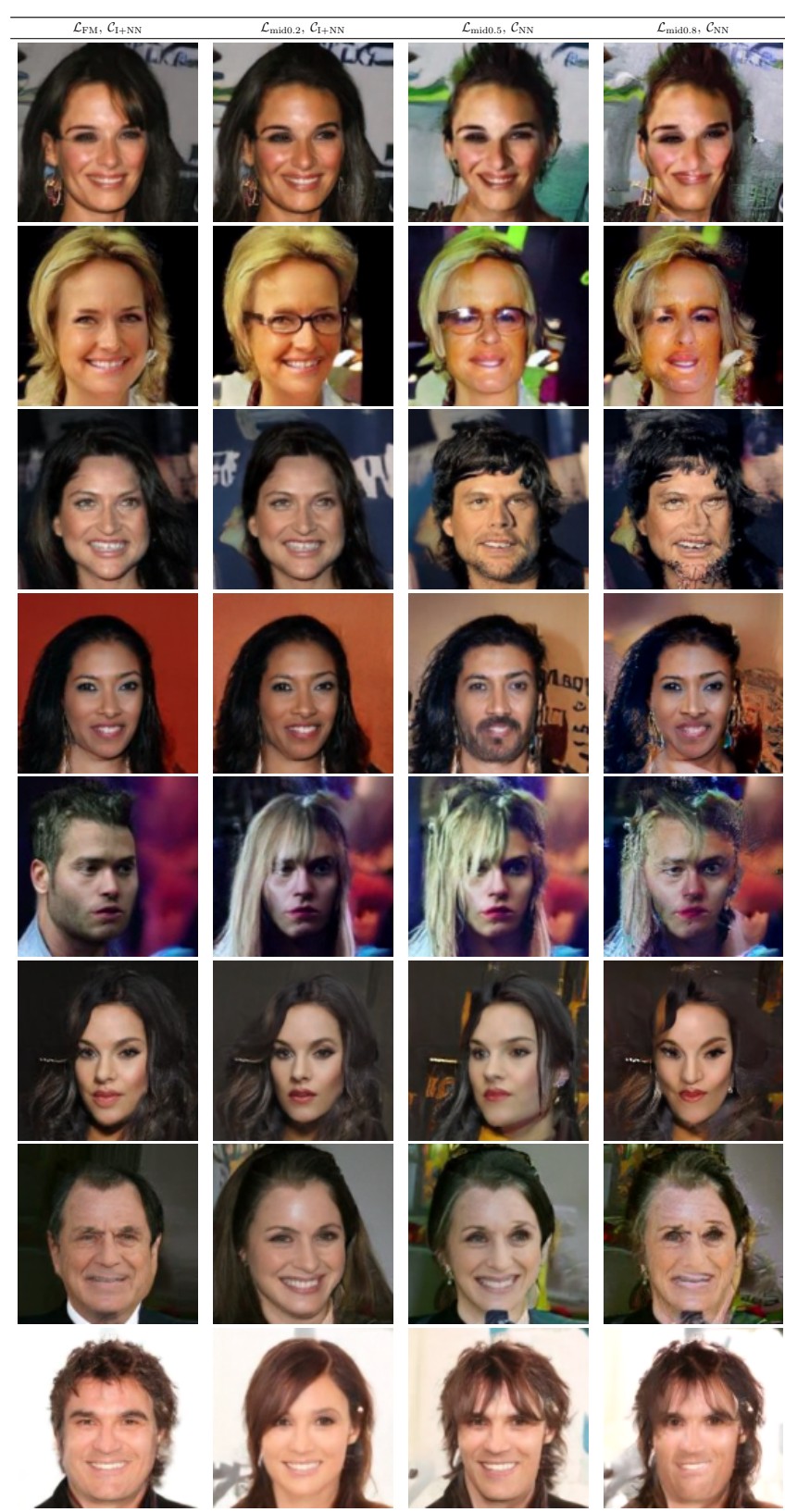

Figure 25: CelebA $128 \times 128$ samples generated by models trained with different *mid weightings*. Each row corresponds to one fixed initial point $x_0$.

