# OpenReview forum: "The generation phases of Flow Matching: a denoising perspective"
_ICLR.cc/2026/Conference — Submitted to ICLR 2026_

### Official Review · Reviewer_DpMZ · 2025-10-26

**Soundness:** 3
**Presentation:** 3
**Contribution:** 3
**Rating:** 6
**Confidence:** 4

**Summary:**

This paper design a framework to empirically study the generation process of flow matching algorithms. Under their framework, they study a combination of three losses (FM, classic denoising  and unweighted denoising) and two parametrization of neural nets. Besides studying studying common generative and inpainting metrics, they also study 1) drift-type and noise-type pertubations and 2) the influence of early and late times on generation, by doing Jacobian/ Lipschitz analysis and tweaking the weighting in loss.

**Strengths:**

This paper provides a framework to study FM generation empirically. Although many similar observations have been reported in other papers, consolidating these scattered observations into one paper clarifies the landscape and may yield new insights. Besides summary work, the study on  Jacobian/ Lipschitz analysis of vector fields over time provides useful intuition.

**Weaknesses:**

1. The section 3.3 Denoising Losses L unifies different losses by reparametrization, and the results are very similar to Kingma & Gao, 2023 [1]. Although the authors cite the paper in intro, it would be better to emphasize this similarity here.
2. The paper consider 3 losses: FM, classic denoising (classic) and unweighted denoising (den). Under their parametrization, FM loss puts more weight on large $t$ (high SNR region), while classic one puts less weight on small $t$ (low SNR). However, when comparing commonly used FM/ Diffusion losses, the FM is actually very aggresive on low SNR (small $t$) ([1] and [2]), and empirically putting more weight in middle SNR can be more helpful (SD3 did this). In this paper, they only study losses that put even more weight on low SNR than FM. For completeness, it would be great to see the performance weighting in other direction.

[1] Kingma, Gao, “Understanding Diffusion Objectives as the ELBO with Simple Data Augmentation”, Neural Information Processing Systems, 2024.

[2] Gao, R., Hoogeboom, E., Heek, J., De Bortoli, V., Murphy, K. P., & Salimans, T. (2025). Diffusion Models and Gaussian Flow Matching: Two Sides of the Same Coin. In The Fourth Blogpost Track at ICLR 2025. https://openreview.net/forum?id=C8Yyg9wy0s

**Questions:**

In section 6.2, can you provide any intution on why using $w_t^{mid}$ lead to such generation results?

---

> ### Author Response · Authors · 2025-11-23
> **Rebuttal by Authors**
>
> We thank the reviewer for the positive feedback and for highlighting what we consider as two keypoints of our work: the unified denoising framework and the spatial regularity of the velocity.
>
> - *section 3.3 &  Kingma & Gao, 2023*
>
> We added a clearer reference to the work of Kingma & Gao in Section 3.3. We also completed the paper with **a detailed related works section in Appendix A** that discusses different works that design weighting schemes and/or harmonize different parametrizations/weightings design choices.
>
> - *the paper only studies losses that put even more weight on low SNR than FM. For completeness, it would be great to see the performance weighting in other direction.*
>
> Thank you for pointing out the blogpost reference. Indeed, in the $\varepsilon$-prediction setting, the weighting induced by the FM loss rewrites as $1/t^2$ which puts more weight on low SNR (as plotted in the blogpost). In the revised related-works section (Appendix A), we now explicitly discuss this perspective and summarize prior work on alternative weighting schemes.
> The choices we made were taken from the denoising perspective (i.e. how denoisers are usually trained versus. generative models) in order to highlight the opposite times that appear in the weightings. What matters indeed is to consider pairs of weightings and parametrizations. In the denoising perspective, the best performance obtained with the FM weighting $1/(1-t)^2$ is thus quite counter-intuitive as it emphasizes low noise levels, i.e. high SNR.
>
> As you rightly pointed out, we have added an ablation that explores giving more weight to the higher SNR region (in the $\varepsilon$ perspective). Specifically, we include experiments with weightings $w_t = t^2 / (1-t)^2$ and $w_t = t^2 / (1-t)^4$ (so this corresponds to the classical SNR weighting (uniform in the $\varepsilon-$prediction) and a weighting that emphasizes high SNR ($1/(1-t)^2$ in the $\varepsilon$-prediction). **Denoising / generative performance on all tested weights are in Table 2 (Appendix D).** Using a Gaussian toy example, **we now also provide intuition in Appendix J for why placing more emphasis on time steps close to $t=1$ is well motivated**, and how this choice connects to classical results in regression.
>
> | Model                       | FID   |
> |-----------------------------|-------|
> | FM weighting                 | 3.89  |
> | SNR weighting               | 3.71 |
> | High SNR weighting      | 36.66 |
>
> - *In section 6.2, can you provide any intuition on why using w_mid lead to such generation results?*
>
> We **added in Appendix H.2 a broad range of “mid-like” models** with weightings $w_t = 1/ (t^\star - t)^2$ where we test various values of $t^\star$ (times where we want to put more weight in the loss). For each model, we report the average pairwise distance to the standard FM and we show representative samples. We also provide more intuition on the results. Specifically, weightings that significantly alter the denoising performance in the early/mid time regime often produce class-level changes on CIFAR, showing that poor denoising in this temporal phase strongly affects the ODE trajectories while still maintaining reasonable FID (FID 50k train <10). In contrast, weightings that primarily alter the mid–late regime yield much higher FID (>30) remain closer to the standard baseline in terms of generated samples. This is consistent with the observations made in the perturbation experiments.

---

### Official Review · Reviewer_6GWe · 2025-10-27

**Soundness:** 1
**Presentation:** 1
**Contribution:** 1
**Rating:** 2
**Confidence:** 4

**Summary:**

The paper studies flow matching from a denoising perspective and proposes an empirical framework to probe generation via controlled perturbations (labeled as noise vs. drift). It aims to clarify where/when denoisers succeed/fail in the generation process.

**Strengths:**

The paper studies the generalization ability of flow matching models, which is an important problem.

**Weaknesses:**

1. Though this is an empirical paper, the experiments are not well conducted. For example, the FID protocol for evaluating on 10k test images is not standard, and the model appears undertrained, as the best-reported CIFAR-10 FID (9.44) is far from the FM baseline (2.99; Lipman et al) . With such poor absolute quality, differences across variants may reflect training deficit rather than principled phase behavior.

2. The experiments are not well designed. The 10-denoisers experiment in Section 4.1 claims that training a single uniform network requires a specific weighting scheme to obtain good performance. This contradicts the fact that models in Lipman et al. use a default uniform weighting scheme and achieve good performance. The perturbation experiments in Section 5 lack a clear definition and justification of the perturbation types. The assignment of small-checkerboard as “noise-type” and large-patch as “drift-type” lacks a formal definition. The categories feel arbitrary and risk overgeneralization from very limited perturbation choices. One cannot draw meaningful conclusions from the experiments. The Lipschitz constant related analysis in Section 6 at most is some observations, and it would require more rigorous analysis to justify the claim that maintaining a high Lipschitz constant is beneficial. The conclusion in Section 6.2 reports a finding as that models with similar FID can have different generation results. This is neither surprising nor useful, as bad models can fail in many different ways.

3. When discussing the emergence of similar samples across architectures (see line 451), the paper should cite Zhang, Huijie, et al. (2023), “The emergence of reproducibility and generalizability in diffusion models.” Additionally, it is questionable to say that the L_mid model produces samples that differ from those of other models; to me, the global outline remains similar to those of other models. Additionally, when we check the generation in Figure 14, the similarities are even more apparent.

Lipman et al, Flow Matching for Generative Modeling, ICLR 2023

**Questions:**

1. Why is the FID so high? Were the models trained to convergence?
2. FID protocol: Why not use the standard protocol of generating 50k samples and comparing them to the train distribution? Could you re-run with the standard protocol for computing the FID?

---

> ### Author Response · Authors · 2025-11-23
> **Rebuttal by Authors**
>
> We thank reviewer 6GWe for the constructive feedback, which helped us refine and better frame the experiments.
>
> - *On FID computation “Why is the FID so high? Were the models trained to convergence?”  FID protocol: Why not use the standard protocol of generating 50k samples and comparing them to the train distribution? Could you re-run with the standard protocol for computing the FID"*
>
> Initial higher FID values were mainly due to two reasons:
> We reported the FID on the test set, for which we used only 10k generated samples.
> The models were trained with only 400 epochs.
> We have retrained all CIFAR10 models for 1000 epochs and evaluated the FID over 50k samples,  which is standard practice.
> These two changes now lead to **a FID of 3.9** for the FM baseline. All the conclusions made in the paper remain valid. We updated all results accordingly.
>
> - *The 10-denoisers experiment in Section 4.1 claims that training a single uniform network requires a specific weighting scheme to obtain good performance. This contradicts the fact that models in Lipman et al. use a default uniform weighting scheme and achieve good performance.*
>
> If we correctly interpret the reviewer’s concern, what is the “uniform” weighting in Lipman et al. is uniform in the velocity formulation (in the conditional FM loss).
> As we show in Equation (7), the equivalence from velocity training with uniform weight to denoiser training introduces a factor $w_t = 1/(1−t)^2$.
> Thus, Lipman’s uniform velocity weighting corresponds exactly to the *non-uniform FM weighting* (**in denoising**) that we identify as necessary for good performance in the denoiser formulation.
>
>
> The 10-denoisers experiment makes this explicit: even when training separate denoisers on small time intervals with uniform denoiser weighting (unlike Lipman’s velocity-based uniform weighting), we cannot match the PSNR obtained by the standard FM model. This highlights that the FM-specific weighting induced by the velocity formulation is essential for good results (a rather counter-intuitive result as it emphasizes large times, i.e. small noise levels, where the denoising task should be easier).
>
>
> - *The perturbation experiments in Section 5 lack a clear definition and justification of the perturbation types. The assignment of small-checkerboard as “noise-type” and large-patch as “drift-type” lacks a formal definition. The categories feel arbitrary and risk overgeneralization from very limited perturbation choices.*
>
>
> As the reviewer suggested:
> - We changed the terminology to respectively “high-frequency” (e.g. checkboard with small squares) and “low-frequency” (e.g. constant shifts, checkerboard with large squares).
> - To make the analysis more complete, we added a broader family of perturbations defined as $\delta =h∗g_0$ where $g_0$​ is a fixed random Gaussian noise and $h$ is a low-pass filter with varying cutoffs, producing a family of controlled perturbations from low to high frequencies (Figs. 11/13/15 in Appendix G on Celeba 128).
> -  We also include an ablation over more time intervals and added the same experiments on CelebA-128 (high–resolution, see Figs. 4 / 5 in the main), which yields consistent results.

---

> > ### Author Response · Authors · 2025-11-23
> > **Rebuttal by Authors**
> >
> > - *The Lipschitz constant related analysis in Section 6 at most is some observations, and it would require more rigorous analysis to justify the claim that maintaining a high Lipschitz constant is beneficial.*
> >
> > We have updated Section 6 to justify why maintaining a high Lipschitz constant is beneficial.  We now give a precise definition of the intermediate phase (the time region after the closed-form Lipschitz peak), and we explain why the **closed-form denoiser becomes piecewise constant** in this regime: it always outputs a training image. Hence the denoiser  Lipschitz constant in this temporal phase is $0$, which is clearly incompatible with good denoising performance on test data.
> > To support this intuition, we added a schematic illustration Figure 6.a with description of the temporal phases and numerical measurements showing how trained denoisers maintain a non-zero Lipschitz constant in this region, in contrast to the closed-form solution.
> >
> > To confirm this, we added more intervals of Lipschitz regularisations on CIFAR. The results make it clear that penalizing the Lipschitz constant in the intermediate regime clearly alters the generation performance: maintaining a high Lipschitz constant there is beneficial.
> >
> > | Model                       | FID   |
> > |-----------------------------|-------|
> > | Standard FM                 | 3.89  |
> > | Reg on    \([0.1, 0.3]\)    | 6.47  |
> > | Reg on \([0.3, 0.6]\)       | 29.14 |
> > | Reg on \([0.6, 0.9]\)       | 29.58 |
> >
> > - *The conclusion in Section 6.2 reports a finding as that models with similar FID can have different generation results. This is neither surprising nor useful, as bad models can fail in many different ways.*
> >
> > We have reformulated the conclusion of 6.2 per the reviewer’s suggestion.
> >
> > - *When discussing the emergence of similar samples across architectures (see line 451), the paper should cite Zhang, Huijie, et al. (2023), “The emergence of reproducibility and generalizability in diffusion models.” Additionally, it is questionable to say that the L_mid model produces samples that differ from those of other models; to me, the global outline remains similar to those of other models. Additionally, when we check the generation in Figure 14, the similarities are even more apparent.*
> >
> > We added the citation. Besides, **new samples on mid weightings have been added in Figure 24.**
> > We added in Appendix G.2 a broad range of “mid-like” models with weightings $w_t = 1/ (t^\star - t)^2$ where we test various values of $t^\star$ (times where we want to put more weight in the loss). For each model, we report the average pairwise distance to the standard FM and we show representative samples.
> > These additional experiments clarify the behavior of the models. Weightings that significantly  alter the denoising performance in the early/mid time regime often produce class-level changes on CIFAR, showing that poor denoising in this temporal phase strongly affects the ODE trajectories while still maintaining reasonable FID (FID 50k train <10). In contrast, weightings that primarily alter the mid–late regime yield much higher FID (>30)  yet remain closer to the standard baseline in terms of generated samples.

---

### Official Review · Reviewer_4YpZ · 2025-11-01

**Soundness:** 2
**Presentation:** 3
**Contribution:** 1
**Rating:** 2
**Confidence:** 3

**Summary:**

This paper presents an empirical investigation into the generative process of Conditional Flow Matching (CFM) models through the lens of denoising. The authors construct a "denoising toolkit" by establishing a formal duality between vector fields and denoisers. Different denoising loss and parametrizations are considered. Using this framework, the authors analyze the impact of controlled perturbations (drift- and noise-type) at different temporal phases of generation. The key findings are: 1) The choice of loss weighting and parametrization significantly impacts both denoising and generative performance, with the standard FM approach performs best; 2) Perturbations affect the process differently depending on when they are applied, revealing distinct early (drift-sensitive) and late (noise-sensitive) phases.

**Strengths:**

- The idea of constructing of the "denoising toolkit" can be fruitful
- The analysis of perturbations in Section 5 is one of the most interesting points. The distinction between drift-type and noise-type perturbations and their dependence on time is an interesting observation that contributes to a better understanding of Flow Matching dynamics. The statement that similar FID indicators can have different generative behaviors is important.

**Weaknesses:**

- The absence of theoretical justification or any intuition that would lead to an understanding of the numerical results presented. While the empirical work is extensive, the paper falls short of providing a satisfying explanation for its most critical observations:
     * Why does the FM loss weighting $(\frac1{1-t})^2$, which emphasizes easy (low-noise) denoising tasks, yield the best generative models? This is counter-intuitive and demands a deeper hypothesis beyond its empirical success.
     * Why does residual parameterization $(C_1+NN)$ so consistently outperform others? The paper notes this fact but does not discuss its causes (e.g., implicit regularization, simpler optimization landscape). It would seem that, from the point of view of the rather complex architecture of neural networks, both parameterizations are practically equivalent, since they are derived from each other by linear transformation.
     * The discussion of the "intermediate phase" in Section 6, while interesting, remains somewhat phenomenological. What is the nature of the computations happening in this phase that make it so crucial for learned models, as opposed to the closed-form solution?

- The choice of specific perturbation patterns (checkerboard, shift) feels somewhat arbitrary.
- The selection of time intervals for perturbations $[t_{min},t_{max}]$ (length 0.3) and for the ad-hoc denoisers is not well-justified. An ablation on the interval size or a more principled method for defining these phases would strengthen the claims.
- For a paper whose goal is to draw general conclusions about generative models, empirical evaluation is limited to relatively low-resolution datasets (CIFAR-10, CelebA-64). It is crucial to demonstrate that the identified phases and superiority of FM parameterization are preserved in more complex, higher-resolution datasets (e.g., ImageNet-128/256) to ensure that the results obtained are not artifacts of simpler data domains. Moreover, only image datasets are considered. It would be interesting to consider extensions of the method to other types of data that can be effectively generated using FM-like methods.
-  The paper distinguishes between "early" and "late" phases based on perturbations and an "intermediate" phase from the ad-hoc denoiser. However, the boundaries and relationships between these phases are not clearly defined. A more coherent presentation, combining these observations into a single evolution of the generative process, would greatly improve the paper.

**Questions:**

- Please, see the weakness section
- Can you provide a more formal hypothesis or intuition for why the FM loss weighting and the C_1+NN parametrization are so effective?
- Have you conducted any experiments on higher-resolution datasets (e.g., ImageNet)?
- What was the reasoning behind the specific perturbation types (checkerboard, shift)? Were they designed to test specific hypotheses about the model's sensitivity to high-frequency vs. low-frequency errors?
- The "intermediate phase" is central to your conclusion. Can you define it more precisely? Is it a specific time range (e.g., $t \in [0.3, 0.7]$)? How does this phase relate to the early phase identified by the peak in the Lipschitz constant of the closed-form solution?
- If arbitrary multidimensional datasets are considered, rather than just images, are your results transferable to them?

---

> ### Author Response · Authors · 2025-11-23
> **Rebuttal by Authors**
>
> We thank Reviewer 4YpZ for the constructive and detailed feedback and for highlighting the interest of the denoising framework and the perturbation experiments. We found the review positive and hope that our answer will lead to an increase in the reviewer’s grade.
>
>
> - “*More formal hypothesis or intuition for why the FM loss weighting and the C_1+NN parametrization are so effective*”
>
> 1. **Parametrization**
>
> A key property of the $C_{\mathrm{I} + \mathrm{NN}}$ parametrization is that it explicitly enforces the correct boundary condition at $t=1$, i.e.\ $D_t(x) = x$, whereas a generic neural network parametrization does not and must instead learn this behavior implicitly from data. We believe that this structural bias stabilizes training near $t=1$, where errors are the most detrimental (according to our experiments on perturbations), which partly explains the observed superiority of the parametrization $C_{\mathrm{I} + \mathrm{NN}}$. We have **added this interpretation** explicitly in the revised version (Sec. 3.5). We also observe in Appendix~K that this is not only an asymptotic advantage: the entire training procedure appears to be significantly easier.
>
> 2.  **Loss weighting**
>
> We agree with the reviewer that the optimal FM weighting $1/(1-t)^2$ may appear counter-intuitive at first, since one might expect that denoising low-noise inputs should be easier and thus require less emphasis. **We believe highlighting this puzzling fact is already an interesting finding**.
> To interpret this fact, we propose in Appendix J a connection with the idea of reweighting a regression loss by the inverse noise variance,well-established in statistics. In classical heteroscedastic linear regression, if the observation noise has variance $\sigma^2(t)$, then the generalized least-squares (GLS) estimator – which is optimal among all linear unbiased estimators – is obtained by minimizing a weighted quadratic loss with weight $1/\sigma^2(t)$.
> We draw a connection between this principle and the FM denoising objective. In a tractable linear Gaussian toy model where the data distribution is assumed Gaussian, a maximum-likelihood derivation yields an optimal weighting of the form
> \[
> \frac{1}{w(t)} \propto \frac{1}{\tau^2} + \frac{t^2}{(1-t)^2},
> \]
> where $\tau^2$ denotes the variance of the data distribution. This is close to the SNR-type weighting $(t/(1-t))^2$ (which we now show in Appendix D performs as well as the standard weighting $1/(1-t)^2$). The obtained weighting behaves as $1/(1-t)^2$ for $t$ close to $1$.
>
> - *“Have you conducted any experiments on higher-resolution datasets”*
>
> Following the reviewer’s suggestion, we have added experiments on the **CelebA 128x128** dataset (perturbation setup, see Figures 4/5 in the main paper and 14/15 in Appendix G.; additional results coming soon). The conclusions remain unchanged on higher-resolution data. We also plan to include results on TinyImageNet (200 classes, diverse dataset)  in the next update.
>
> - *What was the reasoning behind the specific perturbation types (checkerboard, shift)? Were they designed to test specific hypotheses about the model's sensitivity to high-frequency vs. low-frequency errors?*
>
> The reviewer is right and we have adopted their suggested terminology. Our goal is indeed to draw a link between denoising capability at every time and global generation performance. Based on the receptive field analysis of the trained UNet models provided in [1,2] that evolved from full image to local patches, we conjectured that the early time generation was more influenced by global changes (low-frequency perturbations) while the late phase was more sensible to local ones (high-frequency).
> To strengthen our analysis, we added a broader family of perturbations defined as $ \delta =h∗g_0$ where $g_0$​ is a fixed random noise pattern and $h$ is a low-pass filter with varying cutoffs, producing a family of controlled perturbations from low to high frequencies (Figs. 11/13/15 in Appendix G on Celeba 128).
>
> [1] Mason Kamb and Surya Ganguli. An analytic theory of creativity in convolutional diffusion models. In ICML, 2025.
> [2] Matthew Niedoba, Berend Zwartsenberg, Kevin Murphy, and Frank Wood. Towards a mechanistic explanation of diffusion model generalization. In ICML, 2025.

---

> > ### Author Response · Authors · 2025-11-23
> > **Rebuttal by Authors**
> >
> > - *The "intermediate phase" is central to your conclusion. Can you define it more precisely? Is it a specific time range (e.g. t in [0.3,0.7])?How does this phase relate to the early phase identified by the peak in the Lipschitz constant of the closed-form solution?*
> >
> > We now clearly define these temporal phases *from the closed-form point of view* and then describe how the trained models behave on each phase (see the edits to **section 6.1 and the summary Figure 6.a**):
> >
> > *Very early phase:* where the closed form denoiser outputs the mean of the dataset: this is the best estimate at very high noise levels. Since the target denoiser is constant at $t=0$, its Jacobian norm is zero at these initial times.
> >
> > *Early phase:* the closed-form velocity/denoiser rapidly changes, visible as a peak in the Jacobian norm. This corresponds to the splitting of ODE trajectories.
> >
> > *Intermediate phase:* after some time threshold $\tau$, points $x_t$ only belong to a single cone $\cC^{(i)}$ and the velocity, equal to $\frac{x^{(i)} -x}{1-t}$, varies very smoothly. Equivalently, the denoiser’s output is constant equal to $x^{(i)}$ and its Jacobian norm vanishes.
> >
> > *Late phase:* due to the factor $1 / (1-t)$ in the velocity formula, the Jacobian norm of the target velocity explodes while whose of the target denoiser remains null.
> >
> >
> > - *If arbitrary multidimensional datasets are considered, rather than just images, are your results transferable to them?*
> >
> > We believe that focusing on images is reasonable and in line with the scope of this work, as the vast majority of prior and related studies in generative diffusion and flow-matching methods primarily focus on image data.

---

### Official Review · Reviewer_VDYy · 2025-11-02

**Soundness:** 3
**Presentation:** 3
**Contribution:** 3
**Rating:** 6
**Confidence:** 2

**Summary:**

The paper reinterprets flow matching as a denoising process, formally proving the equivalence between FM velocities and time-dependent denoisers. Building on this link, it introduces a unified denoising framework with new weighting and parameterization schemes to analyze generative dynamics. Controlled perturbation experiments reveal that global and local effects dominate early and late phases of generation respectively, while intermediate times are most influential for learned models. Empirically, residual parameterization improves both denoising PSNR and generative FID, offering practical insights into training FM models.

**Strengths:**

- The paper cleanly derives the MMSE denoiser (velocity identity) and uses it to recast FM training as weighted denoising, unifying multiple losses (FM, classical, unweighted) in one framework. This makes design choices (weights, parameterizations) explicit and comparable.

- It shows new insight on temporal regimes. By comparing Jacobian spectral norms, the paper shows the closed-form target has an early Lipschitz peak (at trajectory “splitting”), which trained models smooth out

- Residual parameterization consistently helps, and FM-style weighting correlates with both better denoising and generation

**Weaknesses:**

- Most results use small image benchmarks (CIFAR-10, CelebA-64), a single ODE solver (dopri5, 100 steps), and closely related U-Net-style architectures with EMA; this limits claims about “phases” to low-resolution image FM under specific integration and training regimes. Stronger evidence would test (higher resolutions and diverse datasets (e.g., ImageNet-256/512), other samplers/step counts (explicit compute-vs-quality curves), and other architectures (non-U-Net backbones, transformer variants)

- The paper argues that intermediate times matter most for learned models and that residual parameterization/weights drive outcomes; however, some comparisons confound factors (e.g., “10-denoisers” train 10 separate models, total optimization budget and effective capacity differ from a single time-conditioned network). Similarly, perturbations are hand-crafted (checkerboards, shifts, a “residual” relaxation) and calibrated to a fixed PSNR drop; while illuminating, they may not reflect realistic failure modes (e.g., learned adversarial drifts or dataset-aligned corruptions), leaving open whether conclusions hold under less synthetic disturbances.

**Questions:**

Do the phase sensitivities (drift-early, noise-late) and the mid-time importance persist on larger datasets/resolutions? Any preliminary evidence beyond CIFAR-10/CelebA-64?

For the “10-denoisers” setup, can you report total train steps, parameter counts, and wall-clock to ensure fair comparisons to a single time-conditioned model? If you equalize compute/capacity, do the PSNR/FID conclusions still hold?

---

> ### Author Response · Authors · 2025-11-23
> **Rebuttal by Authors**
>
> We sincerely thank reviewer VDYy for the positive feedback and for highlighting what we consider as two key points of our analysis: the unified denoising framework and the study of the spatial regularity of the velocity. Below, we answer the specific questions.
>
> - *“test higher resolutions diverse datasets [...] other samplers/step counts (explicit compute-vs-quality curves), and other architectures [...] “Do the phase sensitivities (drift-early, noise-late) and the mid-time importance persist on larger datasets/resolutions? ”*
>
> 1. Following your suggestion, we re-ran some experiments of Sections 5 and 6 (perturbations/mid weightings) **on Celeba (high resolution 128x128)** and observe the same effects (drifts/noising) in the phase sensitivities: see Figures 4/5 in the main paper and 14/15 in Appendix G. Regarding the mid-time importance, we are currently training several mid weightings on Celeba128. We are also training some TinyImagenet models (diverse dataset with 200 classes) and will provide the additional results ASAP. Due to limited resources, we couldn’t train ImageNet-256/512.
>
>
> 2. Regarding the need for different architectures: In all the experiments, we used SOTA Unet architectures with attention and timestep embedding (either from the reference torchcfm library or from facebookresearch/flowmatching). We believe that we would obtain the same results with other architectures : for diffusion, [1] have shown that multiple architectures (DiT, U-ViT DDPM++, NCSN) end up learning the same velocity field (see their Fig. 1 and 9).
>  [1] Towards a Mechanistic Explanation of Diffusion Model Generalization, ICML 2025, Niedoba et al
>
> 3. We provide in Appendix I an ablation on the choice of the solver (Euler versus Dopri) to ensure it does not change the conclusion (Table 4).
>
> - “*perturbations are hand-crafted (checkerboards, shifts, a “residual” relaxation) and calibrated to a fixed PSNR drop; while illuminating, they may not reflect realistic failure modes (e.g., learned adversarial drifts or dataset-aligned corruptions)*”
>
> We agree with the reviewer and have followed their recommendation.
> Our goal is to understand which controllable characteristics of the perturbations the generation process is sensitive to. To this end, we did the following:
> We updated the terminology to align with signal-processing conventions, using the terms **low-frequency and high-frequency perturbations**.
> We **added a broader family of perturbations** defined as $ \delta =h∗g_0$ where $g_0$​ is a fixed random Gaussian noise and $h$ is a low-pass filter with varying cutoffs, producing a family of controlled perturbations from low to high frequencies (Figs. 11/13/15 in Appendix G on Celeba 128).
>
> A priori, we don’t aim at adversarial perturbations, but perturbations with specific characteristics that are physically understandable. If the reviewer has a specific perturbation in mind, or a reference to existing work on this topic, we would be interested.
>
> - *“For the “10-denoisers” setup, can you report total train steps, parameter counts, and wall-clock to ensure fair comparisons to a single time-conditioned model? If you equalize compute/capacity, do the PSNR/FID conclusions still hold?”*
>
> In our initial 10-denoiser experiment, each denoiser was trained with the same architecture and the same number of epochs as the standard FM baseline. This resulted in a total training cost that was 10× higher. Our goal at that stage was not to equalize compute or capacity, but rather to highlight the following surprising point: even when splitting the time interval into ten parts and increasing the training budget by a factor of 10,**we still cannot match FM’s PSNR performance**.
> For completeness, we now additionally train a 10-denoiser setup under an equalized budget: each of the 10 denoisers is trained for 100 epochs (instead of 1000 epochs for the FM baseline). The total training cost is therefore comparable. As expected, this equal-budget 10-denoiser performs worse than the FM baseline, both in terms of PSNR and FID (see Fig. 2).

---

### Author Response · Authors · 2025-11-23
**General answer**

We thank the reviewers for pointing out several strengths of the paper, contributing to a better understanding of Flow Matching dynamics (VDYy).
Namely, Reviewers VDYy and DpMZ highlight that our paper provides a **unified and clean** (VDYy) framework that “*clarifies the landscape*” (DpMZ) for understanding why generative models work. This framework takes the form of a denoising toolkit described as “*fruitful*” by Rev 4YpZ, which makes the crucial role of **design choices**  “*explicit and comparable*” (VDYy) when crafting a generative model.
 Key findings are highlighted by rev VDYy:
- **Parametrizing velocity vs. denoiser**: parametrizing the velocity by the network  instead of the denoiser provides an **implicit regularization**, enforcing the denoiser to be identity at time $t=1$, which is the expected behaviour
- **Flow Matching loss weighting** leads to the best results although it puts the highest weights on what is typically viewed as an easy task: denoising at small noise levels. This is a new counter-intuitive finding.

The reviewers also point out our contributions on **the temporal regimes of generative models**:
- **Novel experiments on the spatial regularity analysis of the velocity over time**, displaying a gap between closed-form and trained models, which gives “*new insight on temporal regimes*” (VDYy) and “*useful intuition*” (DpMZ).
- **Denoiser perturbation analysis**: Our investigation into how perturbations affect denoiser performance over time is “*an interesting observation that contributes to a better understanding of Flow Matching dynamics*.”  We show that low-frequency perturbations and high-frequency ones have different impacts depending on the time at which they are applied, which leads to “*the important statement that similar FID indicators can have different generative behaviors*” (4YpZ).

The reviewers pointed several valid directions of improvements, which we have followed in our revision:
1. (Additional and revised experiments:) updated results on CIFAR10 + different datasets (Celeba128, TinyImagenet incoming) to complete and strengthen previous findings.
2. (Intuition/hypothesis/perspective:) We added more intuition to strengthen understanding of the design choices of parametrizations (in Section 3 + Appendix K) and weightings (Appendix J+ Table 2). We also added a dedicated paragraph + summary plot to describe the temporal phases of the generative process (in Sec. 6.1).
3. (Section 5: Perturbation analysis) We complete the perturbation analysis by adding new types of perturbations and new time intervals.  We also updated the terminology to align with signal-processing conventions, emphasizing the distinction between *low-frequency* and *high-frequency* perturbations.
4. (Section 6: study of the spatial regularity) We complete the ad-hoc denoisers experiments by testing more regularization intervals + weightings schemes (Appendix H).
5. We added a detailed related works section, which notably highlights that most design choices on weighting schemes/parametrisations are heuristic and do not explore how they shape the generation process over time.

We provide more details about those in individual answers to reviewers. All modifications made to the paper are in green.

---

### Author Response · Authors · 2025-12-03
**Overall rebuttal summary**

The reviewers rightfully pointed to several directions for improvement, and we have incorporated all of them in our revision:
- **Higher-resolution experiments:** most initial experiments were on CIFAR-10, we now include CelebA (128x128) and TinyImagenet in Figures 14, 25, and 27. (perturbation study, mid-weightings and Lipschitz constant comparison)

- **SOTA performance on CIFAR-10:** All CIFAR-10 models have been retrained (more epochs) to compute standard FID-50k and now achieve sota performance for the standard FM baseline (FID = 3.9); the updated results confirm the original conclusions.

- **Improved perturbation analysis:** Section 5 now includes additional perturbation types, revealing both low- and high-frequency behaviors of the models.

- **Improved intuition and clarity:** We added further explanation of parametrization choices (Sec. 3 + App. K) and weightings (App. J + Table 2), as well as a new summary figure illustrating the temporal phases of the generative process (Sec. 6.1).


Regarding reviewer 6GWe (rating 2, confidence 4): while we fully respect the review, we believe the assessment of “unsoundness” and poor presentation is disproportionate compared to the detailed evaluations from the other reviewers (all of whom rated presentation as 3). The reviewer’s main concrete request was to retrain CIFAR-10 models and provide FID-50k, which we have now done, without altering the paper’s conclusions.

---

### Meta-Review · Area_Chair_feWV · 2026-01-07

**Summary:**

Reviewers were primarily concerned with the limited experimental scale and initially poor FID scores, suggesting that the models were significantly undertrained for a generative modeling paper. There was a notable lack of theoretical depth explaining why specific empirical choices, like residual parameterization and FM weighting, consistently outperformed standard alternatives. The perturbation experiments were viewed as heuristic or arbitrary, with reviewer 4YpZ questioning if the identified phases were merely artifacts of the simple datasets. Concerns regarding the novelty of the unified framework were also raised, specifically whether the findings would generalize to high-resolution images or diverse non-image datasets.

I agree with the reviewers that this paper lacks experimental rigor, hence I vote for rejection. I would encourage the authors to work on this paper in the next version.

**Reviewer Concerns:**

The authors successfully addressed undertraining concerns by providing SOTA FID scores on CIFAR-10 and adding CelebA-128 results to show the analysis holds at higher resolutions. But these datasets are also relatively simple and not too diverse. They refined the perturbation analysis by adopting signal-processing terminology (low/high-frequency) and clarified the compute-equivalence for the "10-denoisers" experimental setup. Outstanding concerns include the lack of testing on high-complexity datasets like ImageNet and a lack of empirical evidence for the framework’s transferability to non-image domains.

**Reviewer Scores:**

Reviewer VDYy would likely maintain a 6,  due to some additional experiments. Reviewer 4YpZ would maintain 2 - The authors made the transition to frequency-based terminology and provided new intuitive justifications for the weighting and parameterization schemes. But the lack of experimental rigor still exists. Reviewer 6GWe's concerns of FID protocol was clarified, but poor experimental rigor still exists. Reviewer DpMZ would also stay at 6, as the authors provided the exact alternative SNR-weighting ablations.

---

### Decision · Program_Chairs · 2026-01-26

Reject